# Neural Collapse with Normalized Features:
# A Geometric Analysis over the Riemannian Manifold

**Can Yaras**[*]
University of Michigan
cjyaras@umich.edu

**Peng Wang**[*]
University of Michigan
pengwa@umich.edu

**Zhihui Zhu**
Ohio State University
zhu.3440@osu.edu

**Laura Balzano**
University of Michigan
girasole@umich.edu

**Qing Qu**
University of Michigan
qingqu@umich.edu

## Abstract

When training overparameterized deep networks for classification tasks, it has been widely observed that the learned features exhibit a so-called "neural collapse" phenomenon. More specifically, for the output features of the penultimate layer, for each class the within-class features converge to their means, and the means of different classes exhibit a certain tight frame structure, which is also aligned with the last layer's classifier. As feature normalization in the last layer becomes a common practice in modern representation learning, in this work we theoretically justify the neural collapse phenomenon under normalized features. Based on an unconstrained feature model, we simplify the empirical loss function in a multi-class classification task into a nonconvex optimization problem over the Riemannian manifold by constraining all features and classifiers over the sphere. In this context, we analyze the nonconvex landscape of the Riemannian optimization problem over the product of spheres, showing a benign global landscape in the sense that the only global minimizers are the neural collapse solutions while all other critical points are strict saddle points with negative curvature. Experimental results on practical deep networks corroborate our theory and demonstrate that better representations can be learned faster via feature normalization. Code for our experiments can be found at https://github.com/cjyaras/normalized-neural-collapse.

## 1 Introduction

Despite the tremendous success of deep learning in engineering and scientific applications over the past decades, the underlying mechanism of deep neural networks (DNNs) still largely remains mysterious. Towards the goal of understanding the learned deep representations, a recent line of seminal works [1–5] presents an intriguing phenomenon that persists across a range of canonical classification problems during the terminal phase of training. Specifically, it has been widely observed that last-layer features (i.e., the output of the penultimate layer) and last-layer linear classifiers of a trained DNN exhibit simple but elegant mathematical structures, in the sense that

- **(NC1) Variability Collapse:** the individual features of each class concentrate to their class-means.
- **(NC2) Convergence to Simplex ETF:** the class-means have the same length and are maximally distant. In other words, they form a Simplex Equiangular Tight Frame (ETF).

---

[*]Equal contribution.

36th Conference on Neural Information Processing Systems (NeurIPS 2022).

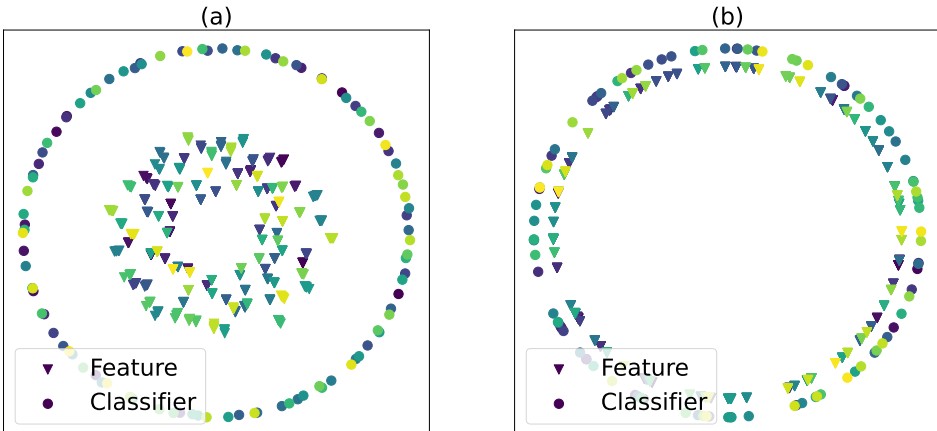

Figure 1: **Comparison of features found with and without normalization.** $K = 100$ classes with $n = 5$ samples per class. Features and classifiers are found through optimizing the cross-entropy loss with a UFM, where features are embedded in 2-dimensional space, i.e., $d = 2$. (a) No normalization of features or classifiers. (b) Features and classifiers are constrained to the unit sphere (features are scaled down for visualization purposes).

Table 1: Average UFM feature loss and accuracy over 10 trials with and without normalization to sphere, with the same set-up as in Figure 1.

|  | **Average CE Loss** | **Average Accuracy** |
|---|---|---|
| No Normalization | $1.63 \pm 0.03$ | $49.9\% \pm 2.39\%$ |
| Normalization | $3.84 \pm 0.00$ | $100.0\% \pm 0.00\%$ |

- **(NC3) Convergence to Self-Duality:** the last-layer linear classifiers perfectly match their class-means.

Such a phenomenon is referred to as *Neural Collapse* ($\mathcal{NC}$) [1], which has been shown empirically to persist across a broad range of canonical classification problems, on different loss functions (e.g., cross-entropy (CE) [1,3,4], mean-squared error (MSE) [5,6], and supervised contrasive (SC) losses [7], etc.), different neural network architectures (e.g., VGG [8], ResNet [9], and DenseNet [10], etc.), and on a variety of standard datasets (e.g., MNIST [11], CIFAR [12], ImageNet [13], etc.). Recently, in independent lines of research, many works are devoted to learning maximally compact and separated features; see, e.g., [14–22]. This has also been widely demonstrated in a number of recent works [23–29], including state-of-the-art natural language models (such as BERT, RoBERTa, and GPT) [24].

**Motivations & contributions.** In this work, we further demystify why $\mathcal{NC}$ happens in network training with feature normalization (i.e., normalizing the last-layer features on the unit hypersphere), mainly motivated by the following reasons:

- *Feature normalization is a common practice in training deep networks.* Recently, many existing results demonstrated that training with feature normalization often improves the quality of learned representation with better class separation [7,16,18–20,30–32]. Such a representation is closely related to the discriminative representation in literature; see, e.g., [16,18,19,33]. As illustrated in Fig. 1 and Table 1, experimental results visualized in low-dimensional space show that features learned with normalization are more uniformly distributed over the sphere and hence are more linearly separable than those learned without normalization. In particular, it has been shown that the learned representations with larger class separation usually lead to improved test performances; see, e.g., [7,34]. Moreover, it has been demonstrated that discriminative representations can also improve robustness to mislabeled data [30,31], and has become a common practice in recent advances on (self-supervised) pre-trained models [32,35].

- *A common practice of theoretically studying $\mathcal{NC}$ with norm constraints.* Due to these practical reasons, many existing theoretical studies on $\mathcal{NC}$ consider formulations with both the norms of features and classifiers constrained [3, 7, 36–38]. Based upon assumptions of unconstrained feature models [3, 4, 39], these works show that the only global solutions satisfy $\mathcal{NC}$ properties for a variety of loss functions (e.g., MSE, CE, SC losses, etc). Nonetheless, they only focused on the global optimality conditions without looking into their nonconvex landscapes, and therefore fail to explain why global $\mathcal{NC}$ solutions can be efficiently reached by classical training algorithms such as stochastic gradient descent (SGD).

In this work, we study the global optimization landscape of the training loss with norm constraints on the features and classifiers. We consider the commonly used CE loss and formulate the problem as a Riemannian optimization problem over products of unit spheres (i.e., the oblique manifold). Our study is also based upon the assumption of the so-called *unconstrained feature model* (UFM) [4, 5, 39] or *layer-peeled model* [40], where the last-layer features of the deep network are treated as free optimization variables to simplify the nonlinear interactions across layers. The underlying reasoning is that modern deep networks are often highly overparameterized with the capacity of learning any representations [41–43], so that the last-layer features can approximate, or interpolate, any point in the feature space.

Assuming the UFM, we show that the Riemannian optimization problem has a benign global landscape, in the sense that the loss with respect to (w.r.t.) the features and classifiers is a strict saddle function [44, 45] over the Riemannian manifold. More specifically, we prove that *every* local minimizer is a global solution satisfying the $\mathcal{NC}$ properties, and *all* the other critical points exhibit directions with negative curvature. Our analysis for the manifold setting is based upon a nontrivial extension of recent studies for the $\mathcal{NC}$ with penalized formulations [2, 4–6], which could be of independent interest. Our work brings new tools from Riemannian optimization for analyzing optimization landscapes of training deep networks with an increasingly common practice of feature normalization. At the same time, we empirically demonstrate the advantages of the constrained formulation on the manifold over its penalized counterpart for training deep networks – faster training and higher quality representations. Lastly, under the UFM we believe the benign landscape over the manifold could hold for many other popular training losses beyond CE, such as the (supervised) contrastive loss [34]. We leave this for future exploration.

**Prior arts and related works on $\mathcal{NC}$.** The empirical $\mathcal{NC}$ phenomenon has inspired a recent line of theoretical studies on understanding why it occurs [4–7, 36, 39, 40]. Like ours, most of these works studied the problem under the UFM. In particular, despite the nonconvexity, recent works showed that the *only* global solutions are $\mathcal{NC}$ solutions for a variety of nonconvex training losses (e.g., CE [4, 36, 40], MSE [5, 6], SC losses [7]) and different problem formulations (e.g., penalized, constrained, and unconstrained) [2, 4–6, 36]. Recently, this study has been extended to deeper models with the MSE training loss [6]. More surprisingly, it has been further shown that the nonconvex losses under the UFM have benign global optimization landscapes, in the sense that every local minimizer satisfies $\mathcal{NC}$ properties and the remaining critical points are strict saddles with negative curvature. Such results have been established for both CE and MSE losses [4, 5], where they considered the unconstrained formulations with regularization on both features and classifiers. We should also mention that the benign global optimization landscapes of many other problems in neural networks have been widely found in the literature; see, e.g., [46–50].

Moreover, there is a line of recent works investigating the benefits of $\mathcal{NC}$ on generalization of deep networks. The work [51] showed that $\mathcal{NC}$ also happens on test data drawn from the same distribution asymptotically, but with less collapse for finite samples [52]. Other works [52, 53] demonstrated that the variability collapse of features is actually happening progressively from shallow to deep layers, and [54] showed that test performance can be improved when enforcing variability collapse on features of intermediate layers. The works [55, 56] showed that fixing the classifier as a simplex ETF improves test performance on imbalanced training data and long-tailed classification problems. For more details on related works, we refer the readers to the Appendix A.

**Notation.** Let $\mathbb{R}^n$ be the $n$-dimensional Euclidean space and $\| \cdot \|_2$ be the Euclidean norm. We write matrices in bold capital letters such as $\boldsymbol{A}$, vectors in bold lower-case such as $\boldsymbol{a}$, and scalars in plain letters such as $a$. Given a matrix $\boldsymbol{A} \in \mathbb{R}^{d \times K}$, we denote its $k$-th column by $\boldsymbol{a}_k$, its $i$-th row by $\boldsymbol{a}^i$, its $(i, j)$-th element by $a_{ij}$, and its spectral norm by $\|\boldsymbol{A}\|$. We use $\text{diag}(\boldsymbol{A})$ to denote a vector

that consists of diagonal elements of $\boldsymbol{A}$, and we use $\mathrm{ddiag}(\boldsymbol{A})$ to denote a diagonal matrix composed by only the diagonal entries of $\boldsymbol{A}$. Given a positive integer $n$, we denote the set $\{1, \ldots, n\}$ by $[n]$. We denote the unit hypersphere in $\mathbb{R}^d$ by $\mathbb{S}^{d-1} := \{\boldsymbol{x} \in \mathbb{R}^d : \|\boldsymbol{x}\|_2 = 1\}$.

## 2   Nonconvex Formulation with Spherical Constraints

In this section, we review the basic concepts of deep neural networks and introduce notation that will be used throughout the paper. Based upon this, we formally introduce the problem formulation over the Riemannian manifold under the assumption of the UFM.

### 2.1   Basics of Deep Neural Networks

In this work, we focus on the multi-class (e.g., $K$ class) classification problem. Given input data $\boldsymbol{x} \in \mathbb{R}^D$, the goal of deep learning is to learn a deep hierarchical representation (or feature) $\boldsymbol{h}(\boldsymbol{x}) = \phi_{\boldsymbol{\theta}}(\boldsymbol{x}) \in \mathbb{R}^d$ of the input along with a linear classifier[1] $\boldsymbol{W} \in \mathbb{R}^{d \times K}$ such that the output $\psi_{\boldsymbol{\Theta}}(\boldsymbol{x}) = \boldsymbol{W}^\top \boldsymbol{h}(\boldsymbol{x})$ of the network fits the input $\boldsymbol{x}$ to an one-hot training label $\boldsymbol{y} \in \mathbb{R}^K$. More precisely, in vanilla form an $L$-layer fully connected deep neural network can be written as

$$\psi_{\boldsymbol{\Theta}}(\boldsymbol{x}) \;=\; \underbrace{\boldsymbol{W}_L}_{\text{linear classifier } \boldsymbol{W} = \boldsymbol{W}_L^\top} \underbrace{\sigma\left(\boldsymbol{W}_{L-1} \cdots \sigma\left(\boldsymbol{W}_1 \boldsymbol{x} + \boldsymbol{b}_1\right) + \boldsymbol{b}_{L-1}\right)}_{\text{feature } \boldsymbol{h} = \phi_{\boldsymbol{\theta}}(\boldsymbol{x})} + \boldsymbol{b}_L, \tag{1}$$

where each layer is composed of an affine transformation, represented by some weight matrix $\boldsymbol{W}_k$, and bias $\boldsymbol{b}_k$, followed by a nonlinear activation $\sigma(\cdot)$, and $\boldsymbol{\Theta} = \{\boldsymbol{W}_k, \boldsymbol{b}_k\}_{k=1}^L$ and $\boldsymbol{\theta} = \{\boldsymbol{W}_k, \boldsymbol{b}_k\}_{k=1}^{L-1}$ denote the weights for *all* the network parameters and those up to the last layer, respectively. Given training samples $\{(\boldsymbol{x}_{k,i}, \boldsymbol{y}_k)\} \subset \mathbb{R}^D \times \mathbb{R}^K$ drawn from the same data distribution $\mathcal{D}$, we learn the network parameters $\boldsymbol{\Theta}$ via minimizing the empirical risk over these samples,

$$\min_{\boldsymbol{\Theta}} \; \sum_{k=1}^K \sum_{i=1}^{n_k} \mathcal{L}_{\mathrm{CE}}\left(\psi_{\boldsymbol{\Theta}}(\boldsymbol{x}_{k,i}), \boldsymbol{y}_k\right), \quad \text{s.t.} \quad \boldsymbol{\Theta} \in \mathcal{C}, \tag{2}$$

where $\boldsymbol{y}_k \in \mathbb{R}^K$ is a one-hot vector with only the $k$th entry being 1 and the remaining ones being 0 for all $k \in [K]$, $\boldsymbol{x}_{k,i} \in \mathbb{R}^D$ is the $i$-th sample in the $k$-th class, $\{n_k\}_{k=1}^K$ denotes the number of training samples in each class, and the set $\mathcal{C}$ denotes the constraint set of the network parameters $\boldsymbol{\Theta}$ that we will specify later.

Here, we study the most widely used CE loss of the form

$$\mathcal{L}_{\mathrm{CE}}(\boldsymbol{z}, \boldsymbol{y}_k) \;:=\; -\log\left(\frac{\exp(z_k)}{\sum_{\ell=1}^K \exp(z_\ell)}\right).$$

### 2.2   Riemannian Optimization over the Product of Spheres

For the $K$-class classification problem, let us consider a simple case where the number of training samples in each class is balanced (i.e., $n = n_1 = n_2 = \cdots = n_K$) and $N = Kn$. We assume that the bias of the final layer $\boldsymbol{b}_L$ is zero with the last activation function $\sigma(\cdot)$ before the output being linear. Analyzing deep networks $\psi_{\boldsymbol{\Theta}}(\boldsymbol{x})$ is a tremendously difficult task mainly due to the *nonlinear interactions* across a large number of layers. To simplify the analysis, we assume the so-called *unconstrained feature model* (UFM) following the previous works [4, 7, 38, 39]. More specifically, we simplify the nonlinear interactions across layers by treating the last-layer features $\boldsymbol{h}_{k,i} = \phi_{\boldsymbol{\theta}}(\boldsymbol{x}_{k,i}) \in \mathbb{R}^d$ as *free* optimization variables, where the underlying reasoning is that modern deep networks are often highly overparameterized to approximate any continuous function [41–43]. Concisely, we write all the features in a matrix form as

$$\boldsymbol{H} = \begin{bmatrix} \boldsymbol{H}_1 & \boldsymbol{H}_2 & \cdots & \boldsymbol{H}_K \end{bmatrix} \in \mathbb{R}^{d \times N}, \; \boldsymbol{H}_k = \begin{bmatrix} \boldsymbol{h}_{k,1} & \boldsymbol{h}_{k,2} & \cdots & \boldsymbol{h}_{k,n} \end{bmatrix} \in \mathbb{R}^{d \times n}, \; \forall \, k \in [K],$$

and correspondingly write the classifier $\boldsymbol{W}$ as

$$\boldsymbol{W} = \begin{bmatrix} \boldsymbol{w}_1 & \boldsymbol{w}_2 & \cdots & \boldsymbol{w}_K \end{bmatrix} \in \mathbb{R}^{d \times K}, \quad \boldsymbol{w}_k \in \mathbb{R}^d, \quad \forall \, k \in [K].$$

---

[1] We write $\boldsymbol{W} = \boldsymbol{W}_L^\top$ in the transposed form for the simplicity of analysis.

Based upon the discussion in Section 1, we assume that both the features $\boldsymbol{H}$ and the classifiers $\boldsymbol{W}$ are normalized,[2] i.e., $\|\boldsymbol{h}_{k,i}\|_2 = 1$ and $\|\boldsymbol{w}_k\|_2 = \tau$ for all $k \in [K]$ and all $i \in [n]$, where $\tau > 0$ is a temperature parameter. As a result, we obtain a *constrained* formulation of the $\mathcal{NC}$ problem over a Riemannian manifold

$$\min_{\boldsymbol{W},\boldsymbol{H}} \ \frac{1}{N}\sum_{k=1}^{K}\sum_{i=1}^{n}\mathcal{L}_{\mathrm{CE}}\left(\boldsymbol{W}^{\top}\boldsymbol{h}_{k,i}, \boldsymbol{y}_k\right) \ \text{s.t.} \ \|\boldsymbol{w}_k\|_2 = \tau, \ \|\boldsymbol{h}_{k,i}\|_2 = 1, \ \forall\, i \in [n], \ \forall\, k \in [K]. \quad (3)$$

Since the temperature parameter $\tau$ can be absorbed into the loss function, we replace $\boldsymbol{w}_k$ by $\tau\boldsymbol{w}_k$ and change the original constraint into $\|\boldsymbol{w}_k\|_2 = 1$ for all $k \in [K]$. Moreover, the product of spherical constraints forms an *oblique manifold* [57] embedded in Euclidean space,

$$\mathcal{OB}(d, K) \ := \ \left\{\boldsymbol{Z} \in \mathbb{R}^{d \times K} \mid \boldsymbol{z}_k \in \mathbb{S}^{d-1}, \ \forall\, k \in [K]\right\}.$$

Consequently, we can rewrite Problem (3) as a Riemannian optimization problem over the oblique manifold w.r.t. $\boldsymbol{W}$ and $\boldsymbol{H}$, i.e.,

$$\min_{\boldsymbol{W},\boldsymbol{H}} \ f(\boldsymbol{W}, \boldsymbol{H}) \ := \ \frac{1}{N}\sum_{k=1}^{K}\sum_{i=1}^{n}\mathcal{L}_{\mathrm{CE}}\left(\tau\boldsymbol{W}^{\top}\boldsymbol{h}_{k,i}, \boldsymbol{y}_k\right), \quad (4)$$
$$\text{s.t.} \quad \boldsymbol{H} \in \mathcal{OB}(d, N), \ \boldsymbol{W} \in \mathcal{OB}(d, K).$$

In Section 3, we will show that all global solutions of this problem satisfy $\mathcal{NC}$ properties, and its objective function is a strict saddle function [58, 59] of $(\boldsymbol{W}, \boldsymbol{H})$ over the oblique manifold so that the $\mathcal{NC}$ solution can be efficiently achieved.

**Riemannian derivatives over the oblique manifold.** In Section 3, we will use tools from Riemannian optimization to characterize the global optimality condition and the geometric properties of the optimization landscape of Problem (4). To proceed, let us first briefly introduce some basic derivations of the Riemannian gradient and Hessian, defined on the tangent space of the oblique manifold. For more technical details, we refer the readers to Appendix B.1. According to [57, Chapter 3 & 5] and [60, 61], we can calculate the Riemannian gradients and Hessian of Problem (4) as follows. Since those quantities are defined on the tangent space, according to [60, Section 3.1] and the illustration in Figure 2, we first have the tangent space to $\mathcal{OB}(d, K)$ at $\boldsymbol{W}$ as

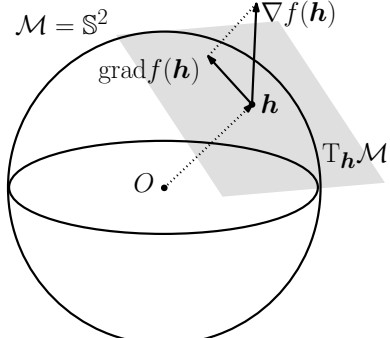

Figure 2: **An illustration of the Riemannian gradient of $f(\boldsymbol{h})$ on a simple manifold $\mathcal{OB}(3, 1) = \mathbb{S}^2$.**

$$\mathrm{T}_{\boldsymbol{W}}\mathcal{OB}(d, K) \ = \ \left\{\boldsymbol{Z} \in \mathbb{R}^{d \times K} \mid \mathrm{diag}\left(\boldsymbol{W}^{\top}\boldsymbol{Z}\right) = \boldsymbol{0}\right\}.$$

This indicates that the tangent space contains all $\boldsymbol{Z}$ such that $\boldsymbol{z}_k$ is orthogonal to $\boldsymbol{w}_k$ for all $k$; when $K = 1$, it reduces to the tangent space to the unit sphere $\mathbb{S}^{d-1}$.

Analogously, we can derive the tangent space for $\boldsymbol{H}$ with a similar form. Let us define

$$\boldsymbol{M} := \tau\boldsymbol{W}^{\top}\boldsymbol{H}, \ g(\boldsymbol{M}) := f(\boldsymbol{W}, \boldsymbol{H}).$$

First, the Riemannian gradient of $f(\boldsymbol{W}, \boldsymbol{H})$ of Problem (4) is basically the projection of the ordinary Euclidean gradient $\nabla f(\boldsymbol{W}, \boldsymbol{H})$ onto its tangent space, i.e., $\mathrm{grad}_{\boldsymbol{W}} f(\boldsymbol{W}, \boldsymbol{H}) = \mathcal{P}_{\mathrm{T}_{\boldsymbol{W}}\mathcal{OB}(d,K)}(\nabla_{\boldsymbol{W}} f(\boldsymbol{W}, \boldsymbol{H}))$ and $\mathrm{grad}_{\boldsymbol{H}} f(\boldsymbol{W}, \boldsymbol{H}) = \mathcal{P}_{\mathrm{T}_{\boldsymbol{H}}\mathcal{OB}(d,N)}(\nabla_{\boldsymbol{H}} f(\boldsymbol{W}, \boldsymbol{H}))$. More specifically, we have

$$\mathrm{grad}_{\boldsymbol{W}} f(\boldsymbol{W}, \boldsymbol{H}) \ = \ \tau\boldsymbol{H}\nabla g(\boldsymbol{M})^{\top} - \tau\boldsymbol{W}\,\mathrm{ddiag}\left(\boldsymbol{W}^{\top}\boldsymbol{H}\nabla g(\boldsymbol{W})^{\top}\right), \quad (5)$$

$$\mathrm{grad}_{\boldsymbol{H}} f(\boldsymbol{W}, \boldsymbol{H}) \ = \ \tau\boldsymbol{W}\nabla g(\boldsymbol{M}) - \tau\boldsymbol{H}\,\mathrm{ddiag}\left(\boldsymbol{H}^{\top}\boldsymbol{W}\nabla g(\boldsymbol{M})\right). \quad (6)$$

Second, for any $\boldsymbol{\Delta} = (\boldsymbol{\Delta}_{\boldsymbol{W}}, \boldsymbol{\Delta}_{\boldsymbol{H}}) \in \mathbb{R}^{d \times K} \times \mathbb{R}^{d \times N}$, we compute the Hessian bilinear form of $f(\boldsymbol{W}, \boldsymbol{H})$ along the direction $\boldsymbol{\Delta}$ by

$$\nabla^2 f(\boldsymbol{W}, \boldsymbol{H})[\boldsymbol{\Delta}, \boldsymbol{\Delta}] \ = \ \nabla^2 g(\boldsymbol{M})\left[\tau\left(\boldsymbol{W}^{\top}\boldsymbol{\Delta}_{\boldsymbol{H}} + \boldsymbol{\Delta}_{\boldsymbol{W}}^{\top}\boldsymbol{H}\right), \tau\left(\boldsymbol{W}^{\top}\boldsymbol{\Delta}_{\boldsymbol{H}} + \boldsymbol{\Delta}_{\boldsymbol{W}}^{\top}\boldsymbol{H}\right)\right]$$
$$+ 2\tau\left\langle\nabla g(\boldsymbol{M}), \boldsymbol{\Delta}_{\boldsymbol{W}}^{\top}\boldsymbol{\Delta}_{\boldsymbol{H}}\right\rangle. \quad (7)$$

---

[2]In practice, it is a common practice to normalize the output feature $\boldsymbol{h}_o$ by its norm, i.e., $\boldsymbol{h} = \boldsymbol{h}_o/\|\boldsymbol{h}_o\|_2$, so that $\|\boldsymbol{h}\|_2 = 1$.

We compute the Riemannian Hessian bilinear form of $f(\boldsymbol{W}, \boldsymbol{H})$ along any direction $\boldsymbol{\Delta} \in \mathrm{T}_{\boldsymbol{W}} \mathcal{OB}(d, K) \times \mathrm{T}_{\boldsymbol{H}} \mathcal{OB}(d, N)$ by

$$\begin{aligned}
\mathrm{Hess}\, f(\boldsymbol{W}, \boldsymbol{H})[\boldsymbol{\Delta}, \boldsymbol{\Delta}] = \nabla^2 f(\boldsymbol{W}, \boldsymbol{H})[\boldsymbol{\Delta}, \boldsymbol{\Delta}] - \langle \boldsymbol{\Delta}_{\boldsymbol{W}}\, \mathrm{ddiag}\left(\boldsymbol{M} \nabla g(\boldsymbol{M})^\top\right), \boldsymbol{\Delta}_{\boldsymbol{W}} \rangle \\
- \langle \boldsymbol{\Delta}_{\boldsymbol{H}}\, \mathrm{ddiag}\left(\boldsymbol{M}^\top \nabla g(\boldsymbol{M})\right), \boldsymbol{\Delta}_{\boldsymbol{H}} \rangle,
\end{aligned} \tag{8}$$

where the extra terms besides $\nabla^2 f(\boldsymbol{W}, \boldsymbol{H})[\boldsymbol{\Delta}, \boldsymbol{\Delta}]$ compensate for the curvature on the oblique manifold. For derivations of (7) and (8), see Appendix B.2. In the following section, we will use the Riemannian gradient and Hessian to characterize the optimization landscape of Problem (4).

## 3   Main Theoretical Analysis

In this section, we first characterize the structure of the global solution set of Problem (4). Based upon this, we analyze the global landscape of Problem (4) via characterizing its Riemannian derivatives.

### 3.1   Global Optimality Condition

For the feature matrix $\boldsymbol{H}$, let us denote the class mean for each class as

$$\overline{\boldsymbol{H}} := \begin{bmatrix} \overline{\boldsymbol{h}}_1 & \cdots & \overline{\boldsymbol{h}}_K \end{bmatrix} \in \mathbb{R}^{d \times K}, \quad \text{where} \quad \overline{\boldsymbol{h}}_k := \frac{1}{n} \sum_{i=1}^{n} \boldsymbol{h}_{k,i}, \ 1 \le k \le K. \tag{9}$$

Based upon this, we show any global solution of Problem (4) exhibits $\mathcal{NC}$ properties in the sense that it satisfies (**NC1**) variability collapse, (**NC2**) convergence to simplex ETF, and (**NC3**) convergence to self-duality.

**Theorem 1** (Global Optimality Condition). *Suppose that the feature dimension is no smaller than the number of classes, i.e., $d \ge K$, and the training labels are balanced in each class, i.e., $n = n_1 = \cdots = n_K$. Then for the CE loss $f(\boldsymbol{W}, \boldsymbol{H})$ in Problem (4), it holds that*

$$f(\boldsymbol{W}, \boldsymbol{H}) \ge \log\left(1 + (K-1)\exp\left(-\frac{K\tau}{K-1}\right)\right)$$

*for all $\boldsymbol{W} = [\boldsymbol{w}_1, \ldots, \boldsymbol{w}_K] \in \mathcal{OB}(d, K)$ and $\boldsymbol{H} = [\boldsymbol{h}_{1,1}, \ldots, \boldsymbol{h}_{K,n}] \in \mathcal{OB}(d, N)$. In particular, equality holds if and only if*

- *(**NC1**) Variability collapse: $\boldsymbol{h}_{k,i} = \overline{\boldsymbol{h}}_k, \ \forall\, i \in [n]$;*

- *(**NC2**) Convergence to Simplex ETF: $\{\overline{\boldsymbol{h}}_k\}_{k=1}^K$ form a sphere-inscribed simplex ETF in the sense that*

$$\overline{\boldsymbol{H}}^\top \overline{\boldsymbol{H}} = \frac{1}{K-1}\left(K \boldsymbol{I}_K - \boldsymbol{1}_K \boldsymbol{1}_K^\top\right), \quad \overline{\boldsymbol{H}} \in \mathcal{OB}(d, K).$$

- *(**NC3**) Convergence to Self-duality: $\boldsymbol{w}_k = \overline{\boldsymbol{h}}_k, \ \forall\, k \in [K]$.*

Compared to the unconstrained regularized problems in [4,5], it is worth noting that the regularization parameters there influence the structure of global solutions, while the temperature parameter $\tau$ only affects the optimization landscape but not the global solutions. On the other hand, the result of our problem (4) is closely related to [7, Theorem 1] (i.e., spherical constraints vs. ball constraints). In fact, our problem and that in [7] share the same global solution set. As such, the proof follows similar ideas of a line of recent works [4–7, 36, 40], and we refer the readers to Appendix C for the proof. It should be noted that we do not claim originality of this result compared to previous works. Instead, our major contribution lies in the following global landscape analysis.

### 3.2   Global Landscape Analysis

Due to the nonconvex nature of Problem (4), the characterization of global optimality alone in Theorem 1 is not sufficient for guaranteeing efficient optimization to those desired global solutions. Thus, we further study the global landscape of Problem (4) by characterizing all the *Riemannian critical points* $(\boldsymbol{W}, \boldsymbol{H}) \in \mathcal{OB}(d, K) \times \mathcal{OB}(d, N)$ satisfying

$$\mathrm{grad}_{\boldsymbol{H}}\, f(\boldsymbol{W}, \boldsymbol{H}) = \boldsymbol{0}, \quad \mathrm{grad}_{\boldsymbol{W}}\, f(\boldsymbol{W}, \boldsymbol{H}) = \boldsymbol{0}.$$

We now state our major result below.

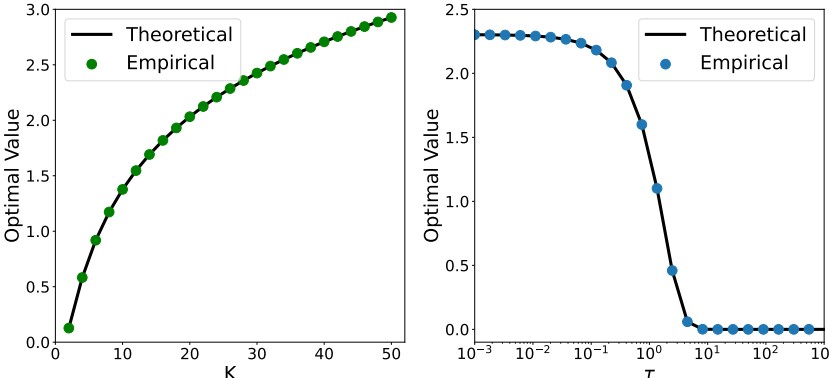

Figure 3: **Global optimization of** (4) **under UFM with** $d = 100$ **and** $n = 5$**.** Theoretical line refers to lower bound (global minimum) from Theorem 1. Empirical values found using gradient descent with random initialization. Left: Lower bound against number of classes $K$ while fixing $\tau = 1$. Right: Lower bound against temperature $\tau$ while fixing $K = 10$. The same empirical values are achieved over many trials due to the benign global landscape.

**Theorem 2** (Global Landscape Analysis). *Assume that the number of training samples in each class is balanced, i.e., $n = n_1 = \cdots = n_K$. If the feature dimension is larger than the number of classes, i.e., $d > K$, and the temperature parameter satisfies $\tau < 2(d-2)(1 + (K \bmod 2)/K)^{-1}$, then the function $f(\boldsymbol{W}, \boldsymbol{H})$ is a strict saddle function that has no spurious local minimum, in the sense that*

- *Any Riemannian critical point $(\boldsymbol{W}, \boldsymbol{H})$ of Problem (4) that is not a local minimizer is a Riemannian strict saddle point with negative curvatures, in the sense that the Riemannian Hessian $\mathrm{Hess} f(\boldsymbol{W}, \boldsymbol{H})$ at the critical point $(\boldsymbol{W}, \boldsymbol{H})$ is non-degenerate, and there exists a direction $\boldsymbol{\Delta} = (\boldsymbol{\Delta_W}, \boldsymbol{\Delta_H}) \in \mathrm{T}_{\boldsymbol{W}} \mathcal{OB}(d, K) \times \mathrm{T}_{\boldsymbol{H}} \mathcal{OB}(d, N)$ such that*

$$\mathrm{Hess} f(\boldsymbol{W}, \boldsymbol{H})[\boldsymbol{\Delta}, \boldsymbol{\Delta}] \; < \; 0.$$

*In other words, $\lambda_{\min} (\mathrm{Hess} f(\boldsymbol{W}, \boldsymbol{H})) < 0$ at the corresponding Riemannian critical point.*

- *Any local minimizer of Problem (4) is a global minimizer of the form shown in Theorem 1.*

For the details of the proof, we refer readers to Appendix D. The second bullet point naturally follows from Theorem 1 and the first bullet point. The major challenge of our analysis is showing the first bullet, i.e., to find a negative curvature direction $\boldsymbol{\Delta}$ for $\mathrm{Hess} f(\boldsymbol{W}, \boldsymbol{H})$. Our key observation is that the set of non-global critical points can be partitioned into two separate cases. In the first case, the last two terms of (8) vanish, and we show that the second term of (7) is negative and dominates the first term for an appropriate direction. We require $\tau$ to not be too large, since the first term is $O(\tau^2)$, whereas the second term is $O(\tau)$. In the second case, using the assumption that $d > K$ we can find a rank-one direction that makes the first term of (7) vanishing. In this case, we similarly show that the second term of (7) is negative but instead dominates the last two terms of (8). In the following, we discuss the implications, relationship, and limitations of our results in Theorem 2.

- *Efficient global optimization to $\mathcal{NC}$ solutions.* Our theorem implies that the $\mathcal{NC}$ solutions can be efficiently reached by Riemannian first-order methods (e.g., Riemannian stochastic gradient descent) with random initialization [58, 62, 63]; see Figure 3 for a demonstration. For training practical deep networks, this can be efficiently implemented by normalizing last-layer features when running SGD.

- *Relation to existing works on $\mathcal{NC}$.* Most existing results have only studied the global minimizers under the UFM [6, 7, 36, 40], which has limited implications for optimization. On the other hand, our landscape analysis is based upon a nontrivial extension of that with the unconstrained problem formulation [4, 5]. Compared to those works, Problem (4) is much more challenging for analysis, due to the fact that the set of critical points of our problem is essentially much larger than that of [4, 5]. Moreover, we empirically demonstrate the advantages of the manifold formulation over its regularized counterpart, in terms of representation quality and training speed.

- *Assumptions on the feature dimension $d$ and temperature parameter $\tau$.* Our current result requires that $d > K$, which is the same as that in [4, 5]. Furthermore, through numerical simulations we conjecture that the global landscape also holds even when $d \ll K$, while the global solutions are uniform over the sphere [36] rather than being simplex ETFs (see Figure 1). The analysis on $d \ll K$ is left for future work. On the other hand, the required upper bound on $\tau$ is for the ease of analysis and it holds generally in practice,[3] but we conjecture that the benign landscape holds without it.

- *Relation to other Riemannian nonconvex problems.* Our result joins a recent line of work on the study of global nonconvex landscapes over Riemannian manifolds, such as orthogonal tensor decomposition [44], dictionary learning [64–68], subspace clustering [69], and sparse blind deconvolution [70–73]. For all these problems constrained over a Riemannian manifold, it can be shown that they exhibit "equivalently good" global minimizers due to symmetries and intrinsic low-dimensional structures, and the loss functions are usually strict saddles [44, 45, 74]. As we can see, the global minimizers (i.e., simplex ETFs) of our problem here also exhibit a similar rotational symmetry, in the sense that $\boldsymbol{W}^\top \boldsymbol{H} = (\boldsymbol{QW})^\top (\boldsymbol{QH})$ for any orthogonal matrix $\boldsymbol{Q}$. Additionally, our result shows that tools from Riemmanian optimization can be powerful for the study of deep learning.

## 4 Experiments

In this section, we support our theoretical results in previous sections and provide further motivation with experimental results on practical deep network training. In the first experiment, we validate the assumption of UFM introduced in Section 2 for analyzing $\mathcal{NC}$, by demonstrating that $\mathcal{NC}$ occurs for increasingly overparameterized deep networks. In the second experiment, we further motivate feature normalization with empirical results, showing that feature normalization can lead to faster training and better collapse than the unconstrained counterpart with regularization. This occurs not only with the UFM but also with practical overparameterized networks. We detail the network architectures, datasets, training details, and metrics used in these experiments, as well as additional experiments in Appendix E.

### 4.1 Validation of the UFM for training networks with feature normalization

In Section 2, our study of the Riemannian optimization problem (4) is based upon the UFM, where we assume $\boldsymbol{H}$ is a free optimization variable. Here, we justify this assumption by showing that $\mathcal{NC}$ happens for training overparameterized networks even when the training labels are completely random. By using random labels, we disassociate the input from their class labels, by which we can characterize the approximation power of the features of overparameterized models. To show this, we train ResNet-18 with varying widths (i.e., the number of feature maps resulting from the first convolutional layer) on CIFAR10 with random labels, with normalized features and classifiers.

As shown in Figure 4, we observe that increasing the width of the network allows for perfect classification on the training data even when the labels are random. Furthermore, increasing the network width also leads to better $\mathcal{NC}$, measured by the decrease in each $\mathcal{NC}$ metric. This validates that (*i*) our assumption of UFM is reasonable given that $\mathcal{NC}$ seems to be independent of the input data, and (*ii*) $\mathcal{NC}$ happens under the constrained formulation (4) on practical networks.

### 4.2 Improved training speed and better representation quality with feature normalization

We now investigate the benefits of using feature normalization, more specifically for improved training speed and better representation quality. First, we consider the UFM formulation, where we optimize Problem (4) and compare to the regularized UFM in [4]. The results are shown in Figure 5. We can see that normalizing features over the sphere consistently results in reaching perfect classification and greater feature collapse (i.e., smaller $\mathcal{NC}_1$) quicker than penalizing the features. To demonstrate that these behaviors are reflected in training practical deep networks, we train both ResNet-18 and ResNet-50 architectures on a reduced CIFAR100 [12] dataset with $N = 3000$ total samples, comparing the training accuracy and metrics of $\mathcal{NC}$ with and without feature normalization. The results are shown in Figure 6.

---

[3]For instance, a standard ResNet-18 [9] model trained on CIFAR-10 [12] has $d = 512$ and $K = 10$. In the same setting, we assume $\tau < 1020$, which is far larger than any useful setting of the temperature parameter (see Appendix E.3).

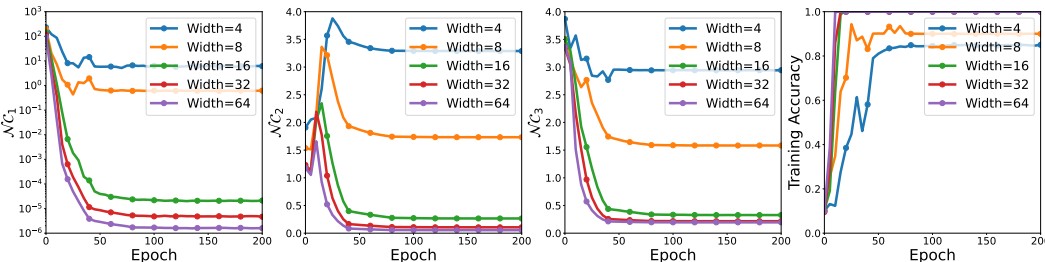

Figure 4: **Validation of UFM on ResNet with varying network width.** $\mathcal{NC}$ metrics and training accuracy of ResNet-18 networks of various widths on CIFAR10 with $n = 200$ over 200 epochs.

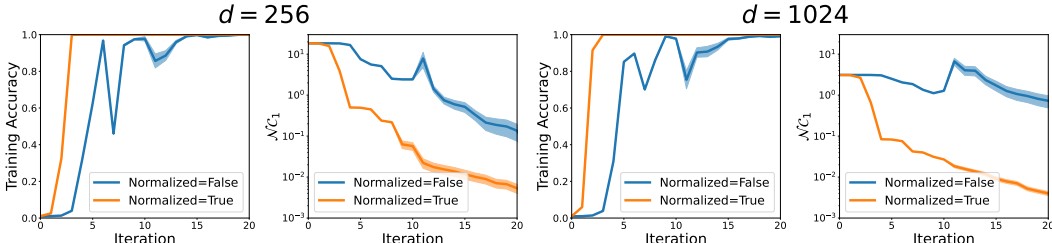

Figure 5: **Faster training/feature collapse of UFM with feature normalization.** Average (deviation denoted by shaded area) training accuracy and $\mathcal{NC}_1$ of UFM over 10 trials of (Riemmanian) gradient descent with backtracking line search. We set $K = 100$ classes, $n = 30$ samples per class.

From Figure 6 (left), we can see that for the ResNet-18 network, we reach perfect classification of the training data about 10-20 epochs sooner by using feature normalization compared to that of the unconstrained formulation. From Figure 6 (right), training the ResNet-50 network without feature normalization for 100 epochs shows slow convergence with poor training accuracy, whereas using feature normalization arrives at above 90% training accuracy in the same number of epochs. By keeping the size of the dataset the same and increasing the number of parameters, it is reasonable that optimizing the ResNet-50 network is more challenging due to the higher degree of overparameterization, yet this effect is mitigated by using feature normalization.

At the same time, for both architectures, using feature normalization leads to greater feature collapse (i.e., smaller $\mathcal{NC}_1$) compared to that of the unconstrained counterpart. As shown in recent work [1, 5, 51] , better $\mathcal{NC}$ often leads to better generalization performance. Experimental results demonstrating that feature normalization generalizes better than regularization are provided in Appendix E.2. Last but not least, we conjecture that the benign landscape holds beyond the CE loss (e.g., SC [7], see Appendix E.4); and we believe the benefits of feature normalization are not limited to the evidence that we showed here, as it could also lead to better robustness [30, 31] that is worth further exploration.

## 5    Conclusion & Discussion

In this work, we study the prevalence of the $\mathcal{NC}$ phenomenon with normalized features. Based upon the assumption of UFM, we formulate the problem as a Riemannian optimization problem over the product of spheres. We showed that the loss function is a strict saddle function over the manifold with respect to the last-layer features and classifiers, with no other spurious local minimizers. We demonstrate this on practical deep network training, and show practical benefits of feature normalization in terms of training speed and learned representation quality. As future work, we would like to expand the study to other popular loss functions such as contrastive loss and study the settings $d \ll K$, which could be of great importance for studying self-supervised learning.

## Acknowledgement

Can Yaras and Qing Qu acknowledge support from U-M START & PODS grants, NSF CAREER CCF 2143904, NSF CCF 2212066, NSF CCF 2212326, and ONR N00014-22-1-2529 grants. Peng Wang and Laura Balzano acknowledge support from ARO YIP W911NF1910027, AFOSR YIP

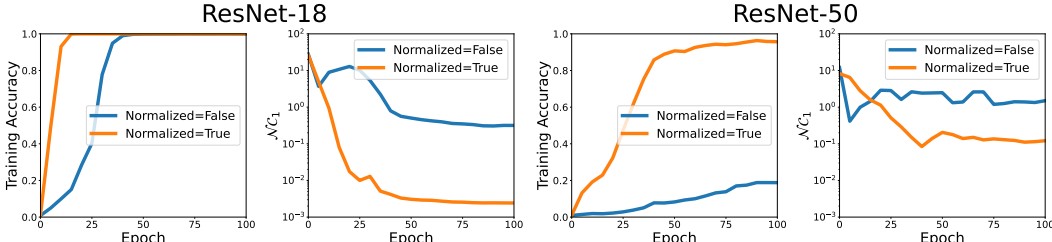

Figure 6: **Faster training/feature collapse with ResNet on CIFAR100 with feature normalization.** Training accuracy and $\mathcal{NC}_1$ of ResNet-18 and ResNet-50 on CIFAR100 with $n = 30$ over 100 epochs.

FA9550-19-1-0026, and NSF CAREER CCF-1845076. Zhihui Zhu acknowledges support from NSF grants CCF-2240708 and CCF-2241298. We would like to thank Zhexin Wu (ETH), Yan Wen (Tsinghua), and Pengru Huang (UMich) for fruitful discussion through various stages of the work.

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
