# Appendix

**Organization of the appendices.**    The appendix is organized as follows. In Appendix A, we first discuss the relationship of our work to prior arts. In Appendix B, we provide some preliminary tools for analyzing our manifold optimization problem. Based upon this, the proof of Theorem 1 and the proof of Theorem 2 are provided in Appendix C and Appendix D, respectively. Finally, our experimental setup as well as more experimental results are provided in Appendix E.

**Notations.**    Before we proceed, let us first introduce the notations that will be used throughout the appendix. Let $\mathbb{R}^n$ denote $n$-dimensional Euclidean space and $\|\cdot\|_2$ be the Euclidean norm. We write matrices in bold capital letters such as $\boldsymbol{A}$, vectors in bold lower-case such as $\boldsymbol{a}$, and scalars in plain letters such as $a$. Given a matrix $\boldsymbol{A} \in \mathbb{R}^{d \times K}$, we denote by $\boldsymbol{a}_k$ its $k$-th column, $\boldsymbol{a}^i$ its $i$-th row, $a_{ij}$ its $(i,j)$-th element, and $\|\boldsymbol{A}\|$ its spectral norm. We use $\mathrm{diag}(\boldsymbol{A})$ to denote a vector that consists of diagonal elements of $\boldsymbol{A}$ and $\mathrm{ddiag}(\boldsymbol{A})$ to denote a diagonal matrix whose diagonal elements are the diagonal ones of $\boldsymbol{A}$. We use $\mathrm{diag}(\boldsymbol{a})$ to denote a diagonal matrix whose diagonal is $\boldsymbol{a}$. Given a positive integer $n$, we denote by $[n]$ the set $\{1, \ldots, n\}$. We denote the unit sphere in $\mathbb{R}^d$ by $\mathbb{S}^{d-1} := \{\boldsymbol{x} \in \mathbb{R}^d : \|\boldsymbol{x}\|_2 = 1\}$.

## A   Related Works

In Section 1, we only provided a brief discussion of related works due to space limitations. In the following, we discuss those related works in more detail. We also refer interested readers to a recent survey on this emerging topic [75].

**The empirical phenomena of $\mathcal{NC}$ and feature engineering.**    Although the seminal works [1] and [2] are the first to summarize the empirical prevalence of $\mathcal{NC}$ for commonly used CE and MSE losses respectively, the idea of designing features with intra-class compactness and inter-class separability has a richer history. More specifically, in the past many loss functions, such as center loss [14], large-margin softmax (L-Softmax) loss [15], and its variants [16, 18–20] are designed with similar goals for the task of visual face recognition. Moreover, the works [16, 18–20] firstly introduced feature normalization and demonstrated its advantages for learning more separable/discriminative features, which well motivates our study in this work. Additionally, related works [21, 22] introduce similar ideas of learning maximal separable features by fixing the linear classifiers with a simplex-shaped structure.

If both the training and test data are drawn from the same distribution, the work [51] shows that $\mathcal{NC}$ also happens on test data asymptotically, but less collapse for finite samples [52]. Under the same setting, the work [5] demonstrated that better collapse could potentially lead to better generalization. However, learning neural collapsed features could easily lead to overfitting [76] and vulnerability to data corruptions [4]. Additionally, the collapse of the feature dimension could cause the loss of intrinsic structure of input data, making the learned features less transferable [77]. In contrast, a line of recent work proposed to learn diverse while discriminative representation by designing a loss that maximizes the coding rate reduction [31, 78]. Instead of collapsing the features to a single dimension, the works promote within-class diversity while keeping the maximum between-class separability. As such, it leads to better robustness and transferability. On the other hand, in self-supervised learning, recent works promote feature diversity and uniformity via contrastive learning [32, 35].

**Global optimality of $\mathcal{NC}$ under UFM.**    The seminal works [1, 2] inspired a lot of recent theoretical studies of the $\mathcal{NC}$ phenomenon. Because the training loss of a deep neural network is highly nonlinear, most works simplify the analysis by assuming unconstrained feature models (UFM) [4, 39] or layer peeled models [40]. It basically assumes that the network has infinite expression power so that the features can be reviewed as free optimization variables. Based upon the UFM, [36] is the first work justifying the global optimality of $\mathcal{NC}$ and uniformity based upon a CE loss with normalized features, while their study is quite simplified in that they assume each class only has one training sample. The work [40] provided global optimality analysis for the CE loss with constrained features under

more generic settings, and they also studies the case when the training samples are imbalanced in each class. The follow-up work [38] extended the analysis to the unconstrained setting without any penalty. Additionally, motivated by the commonly used weight decay on network parameters, the work [4] justifies the global optimality of $\mathcal{NC}$ for the CE loss under unconstrained formulation, with penalization on both the features and classifiers. Its companion work [5] extended the analysis to the MSE loss. Under the same assumption, other work [7] studies the SC loss with normalized features, proving that the only global solutions satisfy $\mathcal{NC}$ properties. Moreover, the work [6] studies the setting beyond the simple UFM, showing that, even for a three-layer nonlinear network, the $\mathcal{NC}$ solutions are the only global solutions with the MSE training loss.

**Benign global landscape and learning dynamics under UFM.**  However, since the training loss is highly nonconvex even under the UFM, merely studying global optimality is not sufficient for guaranteeing efficient global optimization. More recent works address this issue by investigating the global landscape properties and learning dynamics of specific training algorithms. More specifically, under the UFM, [4, 5] showed that the optimization landscapes of CE and MSE losses have benign global optimization landscapes, in the sense that every local minimizer satisfies $\mathcal{NC}$ properties and the remaining critical points are strict saddles with negative curvatures. These works considered the unconstrained formulations with regularization on both features and classifiers. In comparison, our work studies the benign global landscape with features and classifiers constrained over the product of spheres. On the other hand, there is another line of works studying the implicit bias of learning dynamics under UFM [38, 39, 79–82], showing that the convergent direction is along the direction of the minimum-norm separation problem for both CE and MSE losses.

# B  Preliminaries

In this section, we first review some basic aspects of the Riemannian optimization and then compute the derivative of the CE loss.

## B.1  Riemannian Derivatives

According to [57, Chapter 3 & 5] and [60, 61], the tangent space of a general manifold $\mathcal{M} \subseteq \mathbb{R}^d$ at $\boldsymbol{x}$, denoted by $\mathrm{T}_{\boldsymbol{x}}\mathcal{M}$, is defined as the set of all vectors tangent to $\mathcal{M}$ at $\boldsymbol{x}$. Based on this, the Riemannian gradient $\operatorname{grad} f$ of a function $f$ at $\boldsymbol{x}$ is a unique vector in $\mathrm{T}_{\boldsymbol{x}}\mathcal{M}$ satisfying

$$\langle \operatorname{grad} f, \boldsymbol{\xi} \rangle \;=\; Df(\boldsymbol{x})[\boldsymbol{\xi}], \quad \forall\, \boldsymbol{\xi} \in \mathrm{T}_{\boldsymbol{x}}\mathcal{M}.$$

where $Df(\boldsymbol{x})[\boldsymbol{\xi}]$ is the derivative of $f(\gamma(t))$ at $t = 0$, $\gamma(t)$ is any curve on the manifold that satisfies $\gamma(0) = \boldsymbol{x}$ and $\dot{\gamma}(0) = \boldsymbol{\xi}$. The Riemannian Hessian $\operatorname{Hess} f(\boldsymbol{x})$ is a mapping from the tangent space $\mathrm{T}_{\boldsymbol{x}}\mathcal{M}$ to the tangent space $\mathrm{T}_{\boldsymbol{x}}\mathcal{M}$ with

$$\operatorname{Hess} f(\boldsymbol{x})[\boldsymbol{\xi}] \;=\; \widetilde{\nabla}_{\boldsymbol{\xi}} \operatorname{grad} f(\boldsymbol{x}),$$

where $\widetilde{\nabla}$ is the Riemannian connection. For a function $f$ defined on the manifold $\mathcal{M}$, if it can be extended smoothly to the ambient Euclidean space, we have

$$\operatorname{grad} f(\boldsymbol{x}) \;=\; \mathcal{P}_{\mathrm{T}_{\boldsymbol{x}}\mathcal{M}}\left(\nabla f(\boldsymbol{x})\right),$$
$$\operatorname{Hess} f(\boldsymbol{x})[\boldsymbol{\xi}] \;=\; \mathcal{P}_{\mathrm{T}_{\boldsymbol{x}}\mathcal{M}}\left(D \operatorname{grad} f(\boldsymbol{x})[\boldsymbol{\xi}]\right).$$

where $D$ is the Euclidean differential, and $\mathcal{P}_{\mathrm{T}_{\boldsymbol{x}}\mathcal{M}}$ is the projection on the tangent space $\mathrm{T}_{\boldsymbol{x}}\mathcal{M}$. According to [57, Example 3.18], if $\mathcal{M} = \mathbb{S}^{p-1}$, then the tangent space and projection are

$$\mathrm{T}_{\boldsymbol{x}}\mathbb{S}^{p-1} \;=\; \left\{ \boldsymbol{z} \in \mathbb{R}^p \mid \boldsymbol{x}^{\top}\boldsymbol{z} = 0 \right\}, \quad \mathcal{P}_{\mathrm{T}_{\boldsymbol{x}}\mathbb{S}^{p-1}} \boldsymbol{z} \;=\; (\boldsymbol{I} - \boldsymbol{x}\boldsymbol{x}^{\top})\boldsymbol{z}.$$

Moreover, the oblique manifold $\mathcal{M} = \mathcal{OB}(p,q)$ is a product of $q$ unit spheres, and it is also a smooth manifold embedded in $\mathbb{R}^{p \times q}$, where

$$\mathcal{M} = \mathcal{OB}(p,q) \;=\; \underbrace{\mathbb{S}^{p-1} \times \mathbb{S}^{p-1} \times \cdots \times \mathbb{S}^{p-1}}_{q \text{ times}} \;=\; \left\{ \boldsymbol{Z} \in \mathbb{R}^{p \times q} \mid \operatorname{diag}\left(\boldsymbol{Z}^{\top}\boldsymbol{Z}\right) = \mathbf{1} \right\}.$$

Correspondingly, the tangent space $\mathcal{P}_{\mathrm{T}_{\boldsymbol{X}}\mathcal{OB}(p,q)}$ is

$$\mathrm{T}_{\boldsymbol{X}}\mathcal{OB}(p,q) \;=\; \mathrm{T}_{\boldsymbol{x}_1}\mathbb{S}^{p-1} \times \cdots \times \mathrm{T}_{\boldsymbol{x}_q}\mathbb{S}^{p-1} \;=\; \left\{ \boldsymbol{Z} \in \mathbb{R}^{p \times q} \mid \boldsymbol{x}_i^{\top}\boldsymbol{z}_i = 0,\; 1 \le i \le q \right\},$$
$$=\; \left\{ \boldsymbol{Z} \in \mathbb{R}^{p \times q} \mid \operatorname{diag}\left(\boldsymbol{X}^{\top}\boldsymbol{Z}\right) = \mathbf{0} \right\}$$

and the projection operator $\mathrm{T}_{\boldsymbol{x}_1}\mathbb{S}^{p-1}$ is

$$\mathcal{P}_{\mathrm{T}_{\boldsymbol{X}}\mathcal{OB}(p,q)}(\boldsymbol{Z}) = \left[\left(\boldsymbol{I} - \boldsymbol{x}_1\boldsymbol{x}_1^\top\right)\boldsymbol{z}_1 \quad \cdots \quad \left(\boldsymbol{I} - \boldsymbol{x}_q\boldsymbol{x}_q^\top\right)\boldsymbol{z}_q\right]$$
$$= \boldsymbol{Z} - \boldsymbol{X}\,\mathrm{ddiag}(\boldsymbol{X}^\top\boldsymbol{Z}).$$

## B.2  Derivation of (7) and (8)

We first derive (7). Define the curve

$$\begin{aligned}
\phi(t) :&= f(\boldsymbol{W} + t\boldsymbol{\Delta}_{\boldsymbol{W}}, \boldsymbol{H} + t\boldsymbol{\Delta}_{\boldsymbol{H}}) \\
&= g(\tau(\boldsymbol{W} + t\boldsymbol{\Delta}_{\boldsymbol{W}})^\top(\boldsymbol{H} + t\boldsymbol{\Delta}_{\boldsymbol{H}})) \\
&= g(\tau\boldsymbol{W}^\top\boldsymbol{H} + \tau(\boldsymbol{\Delta}_{\boldsymbol{W}}^\top\boldsymbol{H} + \boldsymbol{W}^\top\boldsymbol{\Delta}_{\boldsymbol{H}})t + \tau\boldsymbol{\Delta}_{\boldsymbol{W}}^\top\boldsymbol{\Delta}_{\boldsymbol{H}}t^2) \\
&= g(\boldsymbol{M} + \boldsymbol{\delta}(t))
\end{aligned}$$

where $\boldsymbol{M} = \tau\boldsymbol{W}^\top\boldsymbol{H}$ and $\boldsymbol{\delta}(t) = \tau(\boldsymbol{\Delta}_{\boldsymbol{W}}^\top\boldsymbol{H} + \boldsymbol{W}^\top\boldsymbol{\Delta}_{\boldsymbol{H}})t + \tau\boldsymbol{\Delta}_{\boldsymbol{W}}^\top\boldsymbol{\Delta}_{\boldsymbol{H}}t^2$ satisfies

$$\dot{\boldsymbol{\delta}}(t) = \tau(\boldsymbol{\Delta}_{\boldsymbol{W}}^\top\boldsymbol{H} + \boldsymbol{W}^\top\boldsymbol{\Delta}_{\boldsymbol{H}}) + 2\tau\boldsymbol{\Delta}_{\boldsymbol{W}}^\top\boldsymbol{\Delta}_{\boldsymbol{H}}t$$
$$\ddot{\boldsymbol{\delta}}(t) = 2\tau\boldsymbol{\Delta}_{\boldsymbol{W}}^\top\boldsymbol{\Delta}_{\boldsymbol{H}}$$

so by chain rule and product rule we have

$$\dot{\phi}(t) = \left\langle \dot{\boldsymbol{\delta}}(t), \nabla g(\boldsymbol{M} + \boldsymbol{\delta}(t)) \right\rangle$$

and

$$\ddot{\phi}(t) = \left\langle \ddot{\boldsymbol{\delta}}(t), \nabla g(\boldsymbol{M} + \boldsymbol{\delta}(t)) \right\rangle + \nabla^2 g(\boldsymbol{M} + \boldsymbol{\delta}(t))[\dot{\boldsymbol{\delta}}(t), \dot{\boldsymbol{\delta}}(t)].$$

Then since $\nabla^2 f(\boldsymbol{W}, \boldsymbol{H})[\boldsymbol{\Delta}, \boldsymbol{\Delta}] = \ddot{\phi}(0)$, we have

$$\begin{aligned}
&\nabla^2 f(\boldsymbol{W}, \boldsymbol{H})[\boldsymbol{\Delta}, \boldsymbol{\Delta}] \\
&= \left\langle \ddot{\boldsymbol{\delta}}(0), \nabla g(\boldsymbol{M} + \boldsymbol{\delta}(0)) \right\rangle + \nabla^2 g(\boldsymbol{M} + \boldsymbol{\delta}(0))[\dot{\boldsymbol{\delta}}(0), \dot{\boldsymbol{\delta}}(0)] \\
&= 2\tau\left\langle \boldsymbol{\Delta}_{\boldsymbol{W}}^\top\boldsymbol{\Delta}_{\boldsymbol{H}}, \nabla g(\boldsymbol{M}) \right\rangle + \nabla^2 g(\boldsymbol{M})[\tau(\boldsymbol{\Delta}_{\boldsymbol{W}}^\top\boldsymbol{H} + \boldsymbol{W}^\top\boldsymbol{\Delta}_{\boldsymbol{H}}), \tau(\boldsymbol{\Delta}_{\boldsymbol{W}}^\top\boldsymbol{H} + \boldsymbol{W}^\top\boldsymbol{\Delta}_{\boldsymbol{H}})]
\end{aligned}$$

giving the result.

Now we derive (8). First, we consider the general case of a function $f$ defined on the oblique manifold $\mathcal{M} = \mathcal{OB}(p,q)$, where $f$ can be smoothly extended to the ambient Euclidean space. We have

$$\mathrm{grad}\,f(\boldsymbol{X}) = \nabla f(\boldsymbol{X}) - \boldsymbol{X}\,\mathrm{ddiag}(\boldsymbol{X}^\top\nabla f(\boldsymbol{X})).$$

Then

$$D\,\mathrm{grad}\,f(\boldsymbol{X})[\boldsymbol{U}] = \lim_{t \to 0}\boldsymbol{\Delta}(t)$$

where

$$\begin{aligned}
&\boldsymbol{\Delta}(t) \\
&= \frac{\mathrm{grad}\,f(\boldsymbol{X} + t\boldsymbol{U}) - \mathrm{grad}\,f(\boldsymbol{X})}{t} \\
&= \frac{\nabla f(\boldsymbol{X} + t\boldsymbol{U}) - (\boldsymbol{X} + t\boldsymbol{U})\,\mathrm{ddiag}((\boldsymbol{X} + t\boldsymbol{U})^\top\nabla f(\boldsymbol{X} + t\boldsymbol{U})) - \nabla f(\boldsymbol{X}) + \boldsymbol{X}\,\mathrm{ddiag}(\boldsymbol{X}^\top\nabla f(\boldsymbol{X}))}{t} \\
&= \frac{\nabla f(\boldsymbol{X} + t\boldsymbol{U}) - \nabla f(\boldsymbol{X})}{t} - \boldsymbol{U}\,\mathrm{ddiag}(\boldsymbol{X}^\top\nabla f(\boldsymbol{X} + t\boldsymbol{U})) - \boldsymbol{X}\,\mathrm{ddiag}(\boldsymbol{U}^\top\nabla f(\boldsymbol{X} + t\boldsymbol{U})) \\
&\quad - \boldsymbol{X}\,\mathrm{ddiag}\left(\boldsymbol{X}^\top\frac{\nabla f(\boldsymbol{X} + t\boldsymbol{U}) - \nabla f(\boldsymbol{X})}{t}\right) - t\,\boldsymbol{U}\,\mathrm{ddiag}(\boldsymbol{U}^\top\nabla f(\boldsymbol{X} + t\boldsymbol{U}))
\end{aligned}$$

so

$$\begin{aligned}
D\,\mathrm{grad}\,f(\boldsymbol{X})[\boldsymbol{U}] = {}&\nabla^2 f(\boldsymbol{X})[\boldsymbol{U}] - \boldsymbol{U}\,\mathrm{ddiag}(\boldsymbol{X}^\top\nabla f(\boldsymbol{X})) \\
&- \boldsymbol{X}\,\mathrm{ddiag}(\boldsymbol{U}^\top\nabla f(\boldsymbol{X})) - \boldsymbol{X}\,\mathrm{ddiag}(\boldsymbol{U}^\top\nabla^2 f(\boldsymbol{X})[\boldsymbol{U}]).
\end{aligned}$$

Now, for $\boldsymbol{U} \in \mathrm{T}_{\boldsymbol{X}}\mathcal{M}$, we have $\mathrm{diag}(\boldsymbol{U}^\top\boldsymbol{X}) = \boldsymbol{0}$ so

$$
\begin{aligned}
\mathrm{Hess}\, f(\boldsymbol{X})[\boldsymbol{U}, \boldsymbol{U}] &= \langle \boldsymbol{U}, \mathrm{Hess}\, f(\boldsymbol{X})[\boldsymbol{U}] \rangle \\
&= \langle \boldsymbol{U}, D \,\mathrm{grad}\, f(\boldsymbol{X}) - \boldsymbol{X} \,\mathrm{ddiag}(\boldsymbol{X}^\top D \,\mathrm{grad}\, f(\boldsymbol{X})) \rangle \\
&= \langle \boldsymbol{U}, D \,\mathrm{grad}\, f(\boldsymbol{X}) \rangle \\
&= \langle \boldsymbol{U}, \nabla^2 f(\boldsymbol{X})[\boldsymbol{U}] \rangle - \langle \boldsymbol{U}, \boldsymbol{U} \,\mathrm{ddiag}(\boldsymbol{X}^\top \nabla f(\boldsymbol{X})) \rangle \\
&= \nabla^2 f(\boldsymbol{X})[\boldsymbol{U}, \boldsymbol{U}] - \langle \boldsymbol{U} \,\mathrm{ddiag}(\boldsymbol{X}^\top \nabla f(\boldsymbol{X})), \boldsymbol{U} \rangle .
\end{aligned}
$$

Now let $f$ be defined as in (4). Since $(\boldsymbol{W}, \boldsymbol{H})$ lies on the product manifold $\mathcal{OB}(d, K) \times \mathcal{OB}(d, N) = \mathcal{OB}(d, K + N)$ which is also an oblique manifold, we can simply use the general result above, i.e., for $\boldsymbol{\Delta} \in \mathrm{T}_{(\boldsymbol{W},\boldsymbol{H})} OB(d, N + K)$,

$$
\begin{aligned}
\mathrm{Hess}\, f(\boldsymbol{W}, \boldsymbol{H})[\boldsymbol{\Delta}, \boldsymbol{\Delta}] = {} & \nabla^2 f(\boldsymbol{W}, \boldsymbol{H})[\boldsymbol{\Delta}, \boldsymbol{\Delta}] - \langle \boldsymbol{\Delta}_{\boldsymbol{W}} \,\mathrm{ddiag}(\boldsymbol{W}^\top \nabla_{\boldsymbol{W}} f(\boldsymbol{W}, \boldsymbol{H})), \boldsymbol{\Delta}_{\boldsymbol{W}} \rangle \\
& - \langle \boldsymbol{\Delta}_{\boldsymbol{H}} \,\mathrm{ddiag}(\boldsymbol{H}^\top \nabla_{\boldsymbol{H}} f(\boldsymbol{W}, \boldsymbol{H})), \boldsymbol{\Delta}_{\boldsymbol{H}} \rangle
\end{aligned}
$$

which gives (8) after substituting the ordinary Euclidean gradient of $f$.

### B.3 Derivatives of CE Loss

Note that the CE loss is of the form

$$
\mathcal{L}_{\mathrm{CE}}(\boldsymbol{z}, \boldsymbol{y}_k) = -\log\left( \frac{\exp(z_k)}{\sum_{\ell=1}^{K} \exp(z_\ell)} \right) = \log\left( \sum_{\ell=1}^{K} \exp(z_\ell) \right) - z_k.
$$

Then, one can verify

$$
\frac{\partial \mathcal{L}_{\mathrm{CE}}(\boldsymbol{z}, \boldsymbol{y}_k)}{\partial z_j} = \begin{cases} \frac{\exp(z_j)}{\sum_{\ell=1}^{K} \exp(z_\ell)}, & j \neq k, \\ \frac{\exp(z_j)}{\sum_{\ell=1}^{K} \exp(z_\ell)} - 1, & j = k, \end{cases}
$$

for all $j \in [K]$. Thus, we have

$$
\nabla \mathcal{L}_{\mathrm{CE}}(\boldsymbol{z}, \boldsymbol{y}_k) = \frac{\exp(\boldsymbol{z})}{\sum_{\ell=1}^{K} \exp(z_\ell)} - \boldsymbol{e}_k = \eta(\boldsymbol{z}) - \boldsymbol{e}_k,
$$

where $\eta(\boldsymbol{z})$ is a softmax function, with

$$
\eta(z_j) := \frac{\exp(z_j)}{\sum_{\ell=1}^{K} \exp(z_\ell)}.
$$

Furthermore, we have

$$
\nabla^2 \mathcal{L}_{\mathrm{CE}}(\boldsymbol{z}, \boldsymbol{y}_k) = \mathrm{diag}(\eta(\boldsymbol{z})) - \eta(\boldsymbol{z})\eta(\boldsymbol{z})^\top.
$$

## C Proof of Theorem 1

In this section, we first simplify Problem (4) by utilizing its structure, then characterize the structure of global solutions of the simplified problem, and finally deduce the struture of global solutions of Problem (4) based on their relationship. Before we proceed, we can first reformulate Problem (4) as follows. Let

$$
\boldsymbol{H} = \begin{bmatrix} \boldsymbol{H}^1 & \boldsymbol{H}^2 & \cdots & \boldsymbol{H}^n \end{bmatrix} \in \mathbb{R}^{d \times N}, \ \boldsymbol{H}^i = \begin{bmatrix} \boldsymbol{h}_{1,i} & \boldsymbol{h}_{2,i} & \cdots & \boldsymbol{h}_{K,i} \end{bmatrix} \in \mathbb{R}^{d \times K}, \ \forall\, i \in [N],
$$

and $\bar{f} : \mathbb{R}^{d \times K} \times \mathbb{R}^{d \times K} \to \mathbb{R}$ be such that

$$
\bar{f}(\boldsymbol{W}, \boldsymbol{Q}) = \frac{1}{K} \sum_{k=1}^{K} \mathcal{L}_{\mathrm{CE}}\left( \tau \boldsymbol{W}^\top \boldsymbol{q}_k, \boldsymbol{y}_k \right). \tag{10}
$$

Then, we can rewrite the objective function of Problem (4) as

$$
f(\boldsymbol{W}, \boldsymbol{H}) = \frac{1}{n} \sum_{i=1}^{n} \bar{f}(\boldsymbol{W}, \boldsymbol{H}^i). \tag{11}
$$

**Lemma 1.** *Suppose that* $(\boldsymbol{W}^*, \boldsymbol{Q}^*)$ *is an optimal solution of*

$$\min_{\boldsymbol{W} \in \mathbb{R}^{d \times K}, \boldsymbol{Q} \in \mathbb{R}^{d \times K}} \bar{f}(\boldsymbol{W}, \boldsymbol{Q}) \quad \text{s.t.} \quad \boldsymbol{Q} \in \mathcal{OB}(d, K), \ \boldsymbol{W} \in \mathcal{OB}(d, K). \tag{12}$$

*Then,* $(\boldsymbol{W}^*, \boldsymbol{H}^*)$ *with* $\boldsymbol{H}^* = [\boldsymbol{Q}^* \quad \boldsymbol{Q}^* \quad \cdots \quad \boldsymbol{Q}^*]$ *is an optimal solution of Problem* (4).

*Proof.* According to (11), we note that

$$\min \left\{ f(\boldsymbol{W}, \boldsymbol{H}) : \ \boldsymbol{H} \in \mathcal{OB}(d, N), \ \boldsymbol{W} \in \mathcal{OB}(d, K) \right\}$$

$$\geq \frac{1}{n} \sum_{i=1}^{n} \min \left\{ \bar{f}(\boldsymbol{W}^i, \boldsymbol{H}^i) : \ \boldsymbol{H}^i \in \mathcal{OB}(d, K), \ \boldsymbol{W}^i \in \mathcal{OB}(d, K) \right\},$$

where equality holds if $(\boldsymbol{W}^i, \boldsymbol{H}^i) = (\boldsymbol{W}^*, \boldsymbol{Q}^*)$ for all $i \in [n]$ and $(\boldsymbol{W}, \boldsymbol{H}) = (\boldsymbol{W}^*, \boldsymbol{Q}^*)$. This, together with the fact that $(\boldsymbol{W}^*, \boldsymbol{Q}^*)$ is an optimal solution of Problem (12), implies the desired result. $\square$

Based on the above lemma, it suffices to consider the global optimality condition of Problem (12).

**Proposition 1.** *Suppose that the feature dimension is no smaller than the number of classes (i.e.,* $d \geq K$*) and the training labels are balanced in each class (i.e.,* $n = n_1 = \cdots = n_K$*). Then, any global minimizer* $(\boldsymbol{W}, \boldsymbol{Q}) \in \mathcal{OB}(d, K) \times \mathcal{OB}(d, K)$ *of Problem* (12) *satisfies*

$$\boldsymbol{Q} = \boldsymbol{W}, \quad \boldsymbol{Q}^T \boldsymbol{Q} = \frac{K}{K-1} \left( \boldsymbol{I}_K - \frac{1}{K} \boldsymbol{1}_K \boldsymbol{1}_K^\top \right). \tag{13}$$

*Proof.* According to [4, Lemma D.5], it holds for all $k \in [K]$ and any $c_1 > 0$ that

$$(1 + c_1)(K-1) \left( \mathcal{L}_{\mathrm{CE}} \left( \tau \boldsymbol{W}^\top \boldsymbol{q}_k, \boldsymbol{y}_k \right) - c_2 \right) \geq \tau \left( \sum_{\ell=1}^{K} \boldsymbol{w}_\ell^\top \boldsymbol{q}_k - K \boldsymbol{w}_k^\top \boldsymbol{q}_k \right),$$

where

$$c_2 = \frac{1}{1 + c_1} \log \left( (1 + c_1)(K-1) \right) + \frac{c_1}{1 + c_1} \log \left( \frac{1 + c_1}{c_1} \right)$$

and the equality holds when $\boldsymbol{w}_i^\top \boldsymbol{q}_k = \boldsymbol{w}_j^\top \boldsymbol{q}_k$ for all $i, j \neq k$ and

$$c_1 = \left( (K-1) \exp \left( \frac{\sum_{\ell=1}^{K} \boldsymbol{w}_\ell^\top \boldsymbol{q}_k - K \boldsymbol{w}_k^\top \boldsymbol{q}_k}{K-1} \right) \right)^{-1}.$$

This, together with (10), implies

$$(1 + c_1)(K-1) \left( \bar{f}(\boldsymbol{W}, \boldsymbol{Q}) - c_2 \right) \geq \frac{\tau}{K} \sum_{k=1}^{K} \left( \sum_{\ell=1}^{K} \boldsymbol{w}_\ell^\top \boldsymbol{q}_k - K \boldsymbol{w}_k^\top \boldsymbol{q}_k \right)$$

$$= \frac{\tau}{K} \left( \sum_{k=1}^{K} \sum_{\ell=1}^{K} \boldsymbol{w}_k^\top \boldsymbol{q}_\ell - K \sum_{k=1}^{K} \boldsymbol{w}_k^\top \boldsymbol{q}_k \right)$$

$$= \tau \sum_{k=1}^{K} \boldsymbol{w}_k^\top \left( \bar{\boldsymbol{q}} - \boldsymbol{q}_k \right),$$

where the first inequality becomes equality when $\boldsymbol{w}_i^\top \boldsymbol{q}_k = \boldsymbol{w}_j^\top \boldsymbol{q}_k$ for all $i, j \neq k$ and all $k \in [K]$ and $\bar{\boldsymbol{q}} = \frac{1}{K} \sum_{\ell=1}^{K} \boldsymbol{q}_\ell$ in the last equality. Note that that $\boldsymbol{u}^\top \boldsymbol{v} \geq -\frac{c_3}{2} \|\boldsymbol{u}\|_2^2 - \frac{1}{2c_3} \|\boldsymbol{v}\|_2^2$ for any $c_3 > 0$, where the equality holds when $c_3 \boldsymbol{u} = -\boldsymbol{v}$. Consequently, it holds for any $c_3 > 0$ that

$$(1 + c_1)(K-1) \left( \bar{f}(\boldsymbol{W}, \boldsymbol{Q}) - c_2 \right) \geq -\tau \sum_{k=1}^{K} \left( \frac{c_3}{2} \|\boldsymbol{w}_k\|_2^2 + \frac{1}{2c_3} \|\bar{\boldsymbol{q}} - \boldsymbol{q}_k\|_2^2 \right)$$

$$= -\frac{\tau}{2} \left( c_3 \sum_{k=1}^{K} \|\boldsymbol{w}_k\|_2^2 + \frac{1}{c_3} \sum_{k=1}^{K} \|\boldsymbol{q}_k\|_2^2 - \frac{K}{c_3} \|\bar{\boldsymbol{q}}\|_2^2 \right)$$

$$\geq -\frac{\tau}{2} \left( c_3 \sum_{k=1}^{K} \|\boldsymbol{w}_k\|_2^2 + \frac{1}{c_3} \sum_{k=1}^{K} \|\boldsymbol{q}_k\|_2^2 \right) = -\frac{\tau}{2} \left( c_3 K + \frac{K}{c_3} \right),$$

where the first inequality becomes equality when $c_3 \boldsymbol{w}_k = \boldsymbol{q}_k - \bar{\boldsymbol{q}}$ for all $k \in [K]$, the second inequality becomes equality when $\bar{\boldsymbol{q}} = \boldsymbol{0}$, and the last equality is due to $\boldsymbol{Q} \in \mathcal{OB}(d, K)$ and $\boldsymbol{W} \in \mathcal{OB}(d, K)$. Thus, we have

$$(1 + c_1)(K - 1)\left(\bar{f}(\boldsymbol{W}, \boldsymbol{Q}) - c_2\right) \geq -\frac{\tau K}{2}\left(c_3 + \frac{1}{c_3}\right),$$

where the equality holds when $\boldsymbol{w}_i^\top \boldsymbol{q}_k = \boldsymbol{w}_j^\top \boldsymbol{q}_k$ for all $i, j \neq k$ and all $k \in [K]$, $c_3 \boldsymbol{w}_k = \boldsymbol{q}_k$ for all $k \in [K]$, and $\sum_{k=1}^K \boldsymbol{q}_k = \boldsymbol{0}$. This, together with $\boldsymbol{Q} \in \mathcal{OB}(d, K)$ and $\boldsymbol{W} \in \mathcal{OB}(d, K)$, implies $c_3 = 1$. Thus, we have $\boldsymbol{q}_k = \boldsymbol{w}_k$ for all $k \in [K]$ and

$$\bar{f}(\boldsymbol{W}, \boldsymbol{Q}) \geq -\frac{\tau K}{(1 + c_1)(K - 1)} + c_2.$$

This further implies that $\sum_{k=1}^K \boldsymbol{w}_k = 0$, $\boldsymbol{w}_i^\top \boldsymbol{w}_k = \boldsymbol{w}_j^\top \boldsymbol{w}_k$ for all $i, j \neq k$ and all $k \in [K]$. Then, it holds that for all $1 \leq k \neq \ell \leq K$ that

$$\langle \boldsymbol{w}_\ell, \boldsymbol{w}_k \rangle = -\frac{1}{K - 1}.$$

These, together with $\boldsymbol{Q} \in \mathcal{OB}(d, K)$ and $\boldsymbol{W} \in \mathcal{OB}(d, K)$, imply (13). $\qquad\square$

*Proof of Theorem 1.* According to (11), Lemma 1, and Proposition 1, the global solutions of Problem (4) take the form of

$$\boldsymbol{h}_{k,i} = \boldsymbol{q}_k, \ \boldsymbol{w}_k = \boldsymbol{q}_k, \ \forall \, k \in [K], \ i \in [N],$$

and

$$\boldsymbol{Q}^T \boldsymbol{Q} = \frac{K}{K - 1}\left(\boldsymbol{I}_K - \frac{1}{K}\boldsymbol{1}_K \boldsymbol{1}_K^\top\right).$$

Based on this and the objective function in Problem (4), the value at an optimal solution $(\boldsymbol{W}^*, \boldsymbol{H}^*)$ is

$$f(\boldsymbol{W}^*, \boldsymbol{H}^*) = \log\left(1 + \frac{(K - 1)\exp\left(-\frac{\tau}{K-1}\right)}{\exp(\tau)}\right) = \log\left(1 + (K - 1)\exp\left(-\frac{K\tau}{K - 1}\right)\right).$$

Then, we complete the proof. $\qquad\square$

## D  Proof of Theorem 2

In this section, we first analyze the first-order optimality condition of Problem (4), then characterize the global optimality condition of Problem (4), and finally prove no spurious local minima and strict saddle point property based on the previous optimality conditions. For ease of exposition, let us denote

$$\boldsymbol{M} := \tau \boldsymbol{W}^\top \boldsymbol{H}, \ g(\boldsymbol{M}) := f(\boldsymbol{W}, \boldsymbol{H}) = \frac{1}{N}\sum_{i=1}^n \sum_{k=1}^K \mathcal{L}_{\mathrm{CE}}(\boldsymbol{m}_{k,i}, \boldsymbol{y}_k). \tag{14}$$

Then we have the gradient

$$\nabla f(\boldsymbol{W}, \boldsymbol{H}) = (\nabla_{\boldsymbol{W}} f(\boldsymbol{W}, \boldsymbol{H}), \nabla_{\boldsymbol{H}} f(\boldsymbol{W}, \boldsymbol{H}))$$

with

$$\nabla_{\boldsymbol{W}} f(\boldsymbol{W}, \boldsymbol{H}) = \tau \boldsymbol{H} \nabla g(\boldsymbol{M})^\top, \ \nabla_{\boldsymbol{H}} f(\boldsymbol{W}, \boldsymbol{H}) = \tau \boldsymbol{W} \nabla g(\boldsymbol{M}), \tag{15}$$

and

$$\nabla g(\boldsymbol{M}) = [\eta(\boldsymbol{m}_{1,1}) \quad \cdots \quad \eta(\boldsymbol{m}_{K,n})] - \boldsymbol{I}_K \otimes \boldsymbol{1}_n^\top, \quad \eta(\boldsymbol{m}) = \frac{\exp(\boldsymbol{m})}{\sum_{i=1}^K \exp(m_i)}. \tag{16}$$

## D.1 First-Order Optimality Condition

By using the tools in Appendix B.1, we can calculate the Riemannian gradient at a given point $(\boldsymbol{W}, \boldsymbol{H}) \in \mathcal{OB}(d, N) \times \mathcal{OB}(d, K)$ as in (6) and (5). Thus, for a point $(\boldsymbol{W}, \boldsymbol{H}) \in \mathcal{OB}(d, N) \times \mathcal{OB}(d, K)$, the first-order optimality condition of Problem (4) is

$$\operatorname{grad}_{\boldsymbol{W}} f(\boldsymbol{W}, \boldsymbol{H}) = \tau \boldsymbol{W} \nabla g(\boldsymbol{M}) - \tau \boldsymbol{H} \operatorname{ddiag}\left(\boldsymbol{H}^\top \boldsymbol{W} \nabla g(\boldsymbol{M})\right) = \boldsymbol{0}, \tag{17}$$

$$\operatorname{grad}_{\boldsymbol{H}} f(\boldsymbol{W}, \boldsymbol{H}) = \tau \boldsymbol{H} \nabla g(\boldsymbol{M})^\top - \tau \boldsymbol{W} \operatorname{ddiag}\left(\boldsymbol{W}^\top \boldsymbol{H} \nabla g(\boldsymbol{W})^\top\right) = \boldsymbol{0}. \tag{18}$$

We denote the set of all critical points by

$$\mathcal{C} := \left\{(\boldsymbol{W}, \boldsymbol{H}) \in \mathcal{OB}(d, K) \times \mathcal{OB}(d, N) \mid \operatorname{grad}_{\boldsymbol{H}} f(\boldsymbol{W}, \boldsymbol{H}) = \boldsymbol{0}, \ \operatorname{grad}_{\boldsymbol{W}} f(\boldsymbol{W}, \boldsymbol{H}) = \boldsymbol{0}\right\}.$$

**Lemma 2.** *Suppose that $\boldsymbol{g}_i \in \mathbb{R}^K$ and $\boldsymbol{g}^k \in \mathbb{R}^N$ denote the $i$-th column and $k$-th row vectors of the matrix*

$$\boldsymbol{G} := \nabla g(\boldsymbol{M}) \in \mathbb{R}^{K \times N},$$

*respectively. Let $\boldsymbol{\alpha} \in \mathbb{R}^K$ and $\boldsymbol{\beta} \in \mathbb{R}^N$ be such that*

$$\alpha_k = \left\langle \boldsymbol{w}_k, \boldsymbol{H} \boldsymbol{g}^k \right\rangle, \forall\, k \in [K], \quad \beta_i = \left\langle \boldsymbol{h}_i, \boldsymbol{W} \boldsymbol{g}_i \right\rangle, \forall\, i \in [N]. \tag{19}$$

*Then it holds for any $(\boldsymbol{W}, \boldsymbol{H}) \in \mathcal{C}$ that*

$$\boldsymbol{H} \boldsymbol{g}^k = \alpha_k \boldsymbol{w}_k, \ \forall\, k \in [K], \quad \boldsymbol{W} \boldsymbol{g}_i = \beta_i \boldsymbol{h}_i, \ \forall\, i \in [N]. \tag{20}$$

*and*

$$|\alpha_k| = \|\boldsymbol{H} \boldsymbol{g}^k\|_2, \ k = 1, \dots K, \quad |\beta_i| = \|\boldsymbol{W} \boldsymbol{g}_i\|_2, \ i = 1, \dots, N. \tag{21}$$

*Proof.* According to (15), we have

$$\boldsymbol{H} \boldsymbol{G}^\top = \begin{bmatrix} \boldsymbol{H} \boldsymbol{g}^1 & \dots & \boldsymbol{H} \boldsymbol{g}^K \end{bmatrix}, \quad \boldsymbol{W} \boldsymbol{G} = \begin{bmatrix} \boldsymbol{W} \boldsymbol{g}_1 & \dots & \boldsymbol{W} \boldsymbol{g}_K \end{bmatrix}$$

Using (19), we can compute

$$\operatorname{ddiag}\left(\boldsymbol{W}^\top \boldsymbol{H} \boldsymbol{G}^\top\right) = \operatorname{diag}(\boldsymbol{\alpha}), \quad \operatorname{ddiag}\left(\boldsymbol{H}^\top \boldsymbol{W} \boldsymbol{G}\right) = \operatorname{diag}(\boldsymbol{\beta})$$

This, together with (17) and (18), implies (20). Since $\|\boldsymbol{w}_k\|_2 = 1$ for all $k \in [K]$ and $\|\boldsymbol{h}_i\|_2 = 1$ for all $i \in [N]$, by

$$\alpha_k^2 = \langle \alpha_k \boldsymbol{w}_k, \boldsymbol{H} \boldsymbol{g}^k \rangle = \left\|\boldsymbol{H} \boldsymbol{g}^k\right\|_2^2, \quad \beta_i^2 = \langle \beta_i \boldsymbol{h}_i, \boldsymbol{W} \boldsymbol{g}_i \rangle = \|\boldsymbol{W} \boldsymbol{g}_i\|_2^2$$

which implies (21). $\qquad \square$

## D.2 Characterization of Global Optimality

According to Theorem 1, it holds that for any global solution $(\boldsymbol{W}, \boldsymbol{H}) \in \mathcal{OB}(d, N) \times \mathcal{OB}(d, K)$ that

$$\boldsymbol{H} = \boldsymbol{W} \otimes \boldsymbol{1}_n^\top, \ \boldsymbol{W}^\top \boldsymbol{W} = \frac{K}{K-1}\left(\boldsymbol{I}_K - \frac{1}{K} \boldsymbol{1}_K \boldsymbol{1}_K^\top\right), \tag{22}$$

where $\otimes$ denotes the Kronecker product.

**Lemma 3.** *Given any critical point $(\boldsymbol{W}, \boldsymbol{H}) \in \mathcal{C}$, let $\boldsymbol{\alpha} \in \mathbb{R}^K$ and $\boldsymbol{\beta} \in \mathbb{R}^N$ be defined as in (19). Then, $(\boldsymbol{W}, \boldsymbol{H})$ is a global solution of Problem (4) if and only if the corresponding $(\boldsymbol{\alpha}, \boldsymbol{\beta})$ satisfies*

$$\alpha_k \le -\sqrt{n}\|\nabla g(\boldsymbol{M})\|, \ \forall\, k \in [K], \quad \beta_i \le -\frac{\|\nabla g(\boldsymbol{M})\|}{\sqrt{n}}, \ \forall\, i \in [N], \tag{23}$$

*where $\boldsymbol{M} = \tau \boldsymbol{W}^\top \boldsymbol{H}$.*

*Proof.* Suppose that $(\boldsymbol{W}, \boldsymbol{H}) \in \mathcal{C}$ is an optimal solution. According to (22), one can verify that

$$\boldsymbol{W}^\top \boldsymbol{H} = \boldsymbol{W}^\top \left(\boldsymbol{W} \otimes \mathbf{1}_n^\top\right) = \frac{K}{K-1}\left(\boldsymbol{I}_K - \frac{1}{K}\mathbf{1}_K\mathbf{1}_K^\top\right) \otimes \mathbf{1}_n^\top.$$

According to this and (14), we can compute

$$\nabla g(\boldsymbol{M}) = \frac{-K \exp\left(-\frac{1}{K-1}\right)}{\exp(1) + (K-1)\exp\left(-\frac{1}{K-1}\right)}\left(\boldsymbol{I}_K - \frac{1}{K}\mathbf{1}_K\mathbf{1}_K^\top\right) \otimes \mathbf{1}_n^\top. \tag{24}$$

This, together with $\alpha_k = \langle \boldsymbol{w}_k, \boldsymbol{H}\boldsymbol{g}^k\rangle$, yields for all $k \in K$,

$$\alpha_k = \langle \boldsymbol{H}^\top \boldsymbol{w}_k, \boldsymbol{g}^k\rangle = \frac{-nK \exp\left(-\frac{1}{K-1}\right)}{\exp(1) + (K-1)\exp\left(-\frac{1}{K-1}\right)}. \tag{25}$$

By the same argument, we can compute for all $i \in [N]$,

$$\beta_i = \frac{-K \exp\left(-\frac{1}{K-1}\right)}{\exp(1) + (K-1)\exp\left(-\frac{1}{K-1}\right)}. \tag{26}$$

According to (24), one can verify

$$\|\nabla g(\boldsymbol{M})\| = \frac{\sqrt{n}K \exp\left(-\frac{1}{K-1}\right)}{\exp(1) + (K-1)\exp\left(-\frac{1}{K-1}\right)}.$$

This, together with (25) and (26), implies (23)

Suppose that a critical point $(\boldsymbol{W}^*, \boldsymbol{H}^*) \in \mathcal{C}$ satisfies (23). Let $\boldsymbol{M}^* = \tau \boldsymbol{W}^{*\top}\boldsymbol{H}^*$ and $\lambda = \|\nabla g(\boldsymbol{M}^*)\|$. According to (21) and the fact that $\|\boldsymbol{w}_k^*\| = 1$ and $\|\boldsymbol{h}_k^*\| = 1$ for all $k = 1, \ldots K$, we have

$$\sum_{k=1}^{K} \alpha_k^{*2} = \|\boldsymbol{H}^*\nabla g(\boldsymbol{M}^*)^\top\|_F^2 \le \|\nabla g(\boldsymbol{M}^*)\|^2 \|\boldsymbol{H}^*\|_F^2 = \lambda^2 N,$$

$$\sum_{i=1}^{N} \beta_i^{*2} = \|\boldsymbol{W}^*\nabla g(\boldsymbol{M}^*)\|_F^2 \le \|\nabla g(\boldsymbol{M}^*)\|^2 \|\boldsymbol{W}^*\|_F^2 = \lambda^2 K.$$

This, together with (23), implies

$$\alpha_k^* = -\sqrt{n}\lambda, \ \forall \, k \in [K], \quad \beta_i^* = -\frac{\lambda}{\sqrt{n}}, \ \forall \, i \in [N]. \tag{27}$$

Then, we consider the following regularized problem:

$$\min_{\boldsymbol{W} \in \mathbb{R}^{d \times K}, \boldsymbol{H} \in \mathbb{R}^{d \times N}} f(\boldsymbol{W}, \boldsymbol{H}) + \frac{\lambda\sqrt{n}}{2}\|\boldsymbol{W}\|_F^2 + \frac{\lambda}{2\sqrt{n}}\|\boldsymbol{H}\|_F^2. \tag{28}$$

According to the fact that $(\boldsymbol{W}^*, \boldsymbol{H}^*)$ is a critical point of Problem (4) and satisfies (27), (17), and (18), we have

$$\begin{cases} \boldsymbol{H}^*\nabla g(\boldsymbol{M}^*)^\top + \lambda\sqrt{n}\boldsymbol{W}^* = \boldsymbol{0}, \\ \boldsymbol{W}^*\nabla g(\boldsymbol{M}^*) + \lambda\boldsymbol{H}^*/\sqrt{n} = \boldsymbol{0}. \end{cases} \tag{29}$$

This, together with the first-order optimality condition of Problem (28), yields that $(\boldsymbol{W}^*, \boldsymbol{H}^*)$ is a critical point of Problem (28). According to [67, Lemma C.4] and $\|\nabla g(\boldsymbol{M}^*)\| = \lambda$, it holds that $(\boldsymbol{W}^*, \boldsymbol{H}^*)$ is an optimal solution of Problem (28). This, together with [67, Theorem 3.1], yields that $(\boldsymbol{W}^*, \boldsymbol{H}^*) \in \mathcal{C}$ satisfies

$$\boldsymbol{H}^* = \boldsymbol{W}^* \otimes \mathbf{1}_n^\top, \ \boldsymbol{W}^{*\top}\boldsymbol{W}^* = \frac{K}{K-1}\left(\boldsymbol{I}_K - \frac{1}{K}\mathbf{1}_K\mathbf{1}_K^\top\right).$$

According to Theorem 1, we conclude that $(\boldsymbol{W}^*, \boldsymbol{H}^*)$ is an optimal solution of Problem (4). Then, we complete the proof. $\qquad\square$

## D.3 Negative Curvature at Saddle Points

**Lemma 4.** *Let $\boldsymbol{\alpha}$ and $\boldsymbol{\beta}$ be defined as in Lemma 2. Then $\sum_{k=1}^{K} \alpha_k = \sum_{i=1}^{N} \beta_i$.*

*Proof.* Given the definition of $\boldsymbol{\alpha}$ and $\boldsymbol{\beta}$ in (19), this follows directly from cyclic property of trace:

$$\sum_{k=1}^{K} \alpha_k = \operatorname{trace}(\boldsymbol{W}^\top \boldsymbol{H} \boldsymbol{G}^\top) = \operatorname{trace}(\boldsymbol{G} \boldsymbol{H}^\top \boldsymbol{W}) = \operatorname{trace}(\boldsymbol{H}^\top \boldsymbol{W} \boldsymbol{G}) = \sum_{i=1}^{N} \beta_i,$$

as desired. $\qquad\square$

**Lemma 5.** *Suppose $(\boldsymbol{W}, \boldsymbol{H})$ is a critical point and there exists $i \in [N]$ such that $\beta_i = 0$. Then there exists $\boldsymbol{w} \in \mathbb{S}^{d-1}$ such that $\boldsymbol{W} = \boldsymbol{w} \boldsymbol{1}_K^\top$. Furthermore, we have $\beta_1 = \ldots = \beta_N = 0$.*

*Proof.* Suppose $nk \leq i < n(k+1)$ for $k \in [K]$ (i.e., $\boldsymbol{h}_i$ has label $y_k$). Thus, we can write each entry of the gradient $\boldsymbol{g}_i$ of the CE loss as

$$g_{i\ell} = \begin{cases} p_{ik} - 1 & \ell = k \\ p_{i\ell} & \ell \neq k \end{cases} \quad \text{where} \quad p_{i\ell} = \frac{\exp(\tau \boldsymbol{w}_\ell^\top \boldsymbol{h}_i)}{\sum_{j=1}^{K} \exp(\tau \boldsymbol{w}_j^\top \boldsymbol{h}_i)}.$$

Since $\exp(\cdot) > 0$ and $K \geq 2$, we have $0 < p_{i\ell} < 1$. Given that $\beta_i = 0$ and $\|\boldsymbol{h}_i\|_2 = 1$, from (20) we know that we must have $\boldsymbol{W} \boldsymbol{g}_i = \boldsymbol{0}$, which further gives

$$g_{ik} \boldsymbol{w}_k + \sum_{\ell \neq k} g_{i\ell} \boldsymbol{w}_\ell = 0.$$

Given $1 - p_{ik} > 0$, equivalently we have

$$\boldsymbol{w}_k = \sum_{\ell \neq k} \frac{p_{i\ell}}{1 - p_{ik}} \boldsymbol{w}_\ell,$$

where $\sum_{\ell \neq k} \frac{p_{i\ell}}{1 - p_{ik}} = 1$ and $p_{i\ell} > 0$ so $\boldsymbol{w}_k$ is a strict convex combination of points $\{\boldsymbol{w}_\ell\}_{\ell \neq k}$ on the unit sphere. But since $\boldsymbol{w}_k$ also lies on the unit sphere, and the convex hull of points on the sphere only intersects with the sphere at $\{\boldsymbol{w}_\ell\}_{\ell \neq k}$, we must have all $\boldsymbol{w}_\ell$ be identical, i.e., $\boldsymbol{w}_1 = \ldots = \boldsymbol{w}_K$. Therefore, we can write $\boldsymbol{W} = \boldsymbol{w}_1 \boldsymbol{1}_K^\top$, and consequently

$$\boldsymbol{W} \boldsymbol{G} = \boldsymbol{w}_1 \boldsymbol{1}_K^\top \boldsymbol{G} = \boldsymbol{0},$$

where the last equality follows from the fact that $\boldsymbol{1}_K^\top \boldsymbol{G} = \boldsymbol{1}_K^\top \nabla g(\boldsymbol{M}) = \boldsymbol{0}$. Thus, given $\beta_i = \langle \boldsymbol{h}_i, \boldsymbol{W} \boldsymbol{g}_i \rangle$, from the above we have $\beta_1 = \ldots = \beta_N = 0$. $\qquad\square$

**Lemma 6.** *For any $\boldsymbol{H} \in \mathcal{OB}(d, N)$ and $\boldsymbol{w} \in \mathbb{S}^{d-1}$, there exists at least one $\boldsymbol{a} \in \mathbb{S}^{d-1}$ such that for any $0 < \tau < 2(d-2)(1 + (K \bmod 2)/K)^{-1}$, we have*

$$\boldsymbol{a}^\top \boldsymbol{w} = 0 \quad \text{and} \quad \|\boldsymbol{H}^\top \boldsymbol{a}\|_2^2 < \Gamma := \frac{2N}{\tau(1 + (K \bmod 2)/K) + 2}. \tag{30}$$

*Proof.* To establish the result, we need to show that there exists a linear subspace $\mathcal{S} \subset \mathbb{R}^d$ with $\dim(\mathcal{S}) \geq 2$ such that for any nonzero $\boldsymbol{z} \in \mathcal{S}$ we have $\|\boldsymbol{H}^\top \boldsymbol{z}\|_2^2 < \Gamma \|\boldsymbol{z}\|_2^2$. Then

$$\dim(\mathcal{S} \cap \mathcal{N}(\boldsymbol{w})) > 0,$$

where $\mathcal{N}(\boldsymbol{w})$ denotes the null space of $\boldsymbol{w}$, so if we choose unit-norm $\boldsymbol{a} \in \mathcal{S} \cap \mathcal{N}(\boldsymbol{w})$, we can obtain the desired results. Let $(\sigma_\ell^2(\boldsymbol{H}), v_\ell)$ denote the $\ell$-th eigenvalue-eigenvector pair of $\boldsymbol{H} \boldsymbol{H}^\top \in \mathbb{R}^{d \times d}$ for $\ell \in [d]$. Given the fact $\boldsymbol{H} \in \mathcal{OB}(d, N)$, it is obvious that

$$\|\boldsymbol{H}\|_F^2 = \sum_{\ell=1}^{d} \sigma_\ell^2(\boldsymbol{H}) = \sum_{j=1}^{N} \|\boldsymbol{h}_j\|_2^2 = N.$$

Now suppose that $\sigma_{d-1}^2(\boldsymbol{H}) \geq \Gamma$. Then we must have

$$N = \sum_{i=1}^{d} \sigma_i^2(\boldsymbol{H}) \geq (d-1)\Gamma = (d-1)\frac{2N}{\tau(1 + (K \bmod 2)/K) + 2}$$

which implies $\tau \geq 2(d-2)(1 + (K \bmod 2)/K)^{-1}$, but this contradicts the assumption on $\tau$. Therefore $\sigma_{d-1}^2(\boldsymbol{H}) < \Gamma$, so we can choose $\mathcal{S} = \text{span}(\{v_{d-1}, v_d\})$, which suffices to give the result by the above argument. $\qquad \square$

We are now ready to show that at any critical point that is not globally optimal, we can find a direction along which the Riemannian Hessian has a strictly negative curvature at this point.

Recall $\boldsymbol{M} := \tau \boldsymbol{W}^\top \boldsymbol{H}$ and $\boldsymbol{G} := \nabla g(\boldsymbol{M})$, as well as the definition of $\boldsymbol{\alpha} \in \mathbb{R}^K$, $\boldsymbol{\beta} \in \mathbb{R}^N$ in (19). As mentioned at the beginning of Appendix C, we can write $\boldsymbol{H}$ as

$$\boldsymbol{H} = \begin{bmatrix} \boldsymbol{H}^1 & \boldsymbol{H}^2 & \cdots & \boldsymbol{H}^n \end{bmatrix} \in \mathbb{R}^{d \times N}, \ \boldsymbol{H}^i = \begin{bmatrix} \boldsymbol{h}_{1,i} & \boldsymbol{h}_{2,i} & \cdots & \boldsymbol{h}_{K,i} \end{bmatrix} \in \mathbb{R}^{d \times K}, \ \forall \, i \in [N].$$

As a final remark, the bilinear form of the Riemannian Hessian in (8) can be written as

$$\text{Hess } f(\boldsymbol{W}, \boldsymbol{H})[\boldsymbol{\Delta}, \boldsymbol{\Delta}] = \nabla^2 f(\boldsymbol{W}, \boldsymbol{H})[\boldsymbol{\Delta}, \boldsymbol{\Delta}] - \tau \sum_{k=1}^{K} \alpha_k \|\boldsymbol{\delta}_{W_k}\|_2^2 - \tau \sum_{i=1}^{N} \beta_i \|\boldsymbol{\delta}_{H_i}\|_2^2 \qquad (31)$$

where $\nabla^2 f(\boldsymbol{W}, \boldsymbol{H})[\boldsymbol{\Delta}, \boldsymbol{\Delta}]$ is given in (7), and $\boldsymbol{\delta}_{W_k}$, $\boldsymbol{\delta}_{H_i}$ are the $k$-th and $i$-th columns of $\boldsymbol{\Delta}_{\boldsymbol{W}}$ and $\boldsymbol{\Delta}_{\boldsymbol{H}}$ respectively.

**Proposition 2.** *Suppose $d > K$ and $\tau < 2(d-2)(1 + (K \bmod 2)/K)^{-1}$. For any critical point $(\boldsymbol{W}, \boldsymbol{H}) \in \mathcal{C}$ that is not globally optimal, there exists $\boldsymbol{\Delta} = (\boldsymbol{\Delta}_{\boldsymbol{W}}, \boldsymbol{\Delta}_{\boldsymbol{H}}) \in \mathrm{T}_{\boldsymbol{W}}\mathcal{OB}(d, K) \times \mathrm{T}_{\boldsymbol{H}}\mathcal{OB}(d, N)$ such that*

$$\text{Hess } f(\boldsymbol{W}, \boldsymbol{H})[\boldsymbol{\Delta}, \boldsymbol{\Delta}] < 0. \qquad (32)$$

*Proof.* We proceed by considering two separate cases for the value of $\boldsymbol{\beta}$: $\beta_i = 0$ for some $i \in [N]$, and $\beta_i \neq 0$ for all $i \in [N]$.

**Case 1:** Suppose $\beta_i = 0$ for some $i \in [N]$. In this case, by Lemma 5, we know that $\boldsymbol{W} = \boldsymbol{w}\boldsymbol{1}_K^\top$ for some $\boldsymbol{w} \in \mathbb{S}^{d-1}$ and $\boldsymbol{\beta} = \boldsymbol{0}$. We have that $\boldsymbol{M} = \tau \boldsymbol{1}_K \boldsymbol{w}^\top \boldsymbol{H}$, and so

$$\boldsymbol{G} = -\frac{1}{N}\begin{bmatrix} \boldsymbol{A} & \cdots & \boldsymbol{A} \end{bmatrix} \in \mathbb{R}^{K \times N}, \quad \boldsymbol{A} = \boldsymbol{I}_K - \frac{1}{K}\boldsymbol{1}_K\boldsymbol{1}_K^\top \in \mathbb{R}^{K \times K}. \qquad (33)$$

For the the $i$-th column of $\boldsymbol{M}$, i.e. $\boldsymbol{m}_i$, we have the Hessian

$$\nabla^2 \mathcal{L}_{\text{CE}}(\boldsymbol{m}_i, \boldsymbol{y}_k) = \frac{1}{K}\boldsymbol{I}_K - \frac{1}{K^2}\boldsymbol{1}_K\boldsymbol{1}_K^\top = \frac{1}{K}\boldsymbol{A} \qquad (34)$$

Using Lemma 6, choose $\boldsymbol{a} \in \mathbb{S}^{d-1}$ satisfying (30). Additionally, choose a vector $\boldsymbol{u} \in \mathbb{R}^K$ with each entry $u_k = (-1)^{k+1}$ (noting that $\sum_k u_k = K \bmod 2$). Now, we construct the negative curvature direction $\boldsymbol{\Delta} = (\boldsymbol{\Delta}_{\boldsymbol{W}}, \boldsymbol{\Delta}_{\boldsymbol{H}})$ as

$$\boldsymbol{\Delta}_{\boldsymbol{W}} = \boldsymbol{a}\boldsymbol{u}^\top, \ \boldsymbol{\Delta}_{\boldsymbol{H}} = \begin{bmatrix} \boldsymbol{\Delta}_{\boldsymbol{H}^1} & \cdots \boldsymbol{\Delta}_{\boldsymbol{H}^n} \end{bmatrix}$$

where

$$\boldsymbol{\Delta}_{\boldsymbol{H}^i} = \boldsymbol{a}\boldsymbol{u}^\top - \boldsymbol{H}^i \, \text{ddiag}(\boldsymbol{H}^{i\top}\boldsymbol{a}\boldsymbol{u}^\top), \ \forall i \in [n].$$

First, let $\boldsymbol{\delta}_{M_i}$ denote the $i$-th column of $\boldsymbol{\Delta}_{\boldsymbol{M}} := \boldsymbol{W}^\top \boldsymbol{\Delta}_{\boldsymbol{H}} + \boldsymbol{\Delta}_{\boldsymbol{W}}^\top \boldsymbol{H}$, so that

$$\boldsymbol{\delta}_{M_i} = (\boldsymbol{w}^\top \boldsymbol{\delta}_{H_i})\boldsymbol{1}_K + (\boldsymbol{h}_i^\top \boldsymbol{a})\boldsymbol{u}. \qquad (35)$$

Then from (14) and (34), we know that

$$\nabla^2 g(\boldsymbol{W}^\top \boldsymbol{H})[\tau\boldsymbol{\Delta}_{\boldsymbol{M}}, \tau\boldsymbol{\Delta}_{\boldsymbol{M}}] = \frac{\tau^2}{NK}\sum_{i=1}^{N} \boldsymbol{\delta}_{M_i}^\top \boldsymbol{A}\boldsymbol{\delta}_{M_i}.$$

Since $\boldsymbol{A}\mathbf{1}_K = \mathbf{0}$ and $\boldsymbol{u}^\top \boldsymbol{A}\boldsymbol{u} = K - (K \bmod 2)/K$, by (35) we have

$$\nabla^2 g(\boldsymbol{W}^\top \boldsymbol{H})\left[\tau\boldsymbol{\Delta}_M, \tau\boldsymbol{\Delta}_M\right] = \frac{\tau^2}{NK}\left(K - \frac{K \bmod 2}{K}\right)\sum_{i=1}^{N}(\boldsymbol{h}_i^\top \boldsymbol{a})^2$$

$$= \frac{\tau^2}{NK}\left(K - \frac{K \bmod 2}{K}\right)\|\boldsymbol{H}^\top \boldsymbol{a}\|_2^2.$$

On the other hand, by (33) we have

$$2\tau\left\langle\boldsymbol{G}, \boldsymbol{\Delta}_W^\top\boldsymbol{\Delta}_H\right\rangle = -\frac{2\tau}{N}\sum_{i=1}^{n}\mathrm{trace}(\boldsymbol{A}\boldsymbol{\Delta}_W^\top\boldsymbol{\Delta}_{H^i})$$

$$= -\frac{2\tau}{N}\sum_{i=1}^{n}\mathrm{trace}\left(\boldsymbol{A}\boldsymbol{u}\boldsymbol{u}^\top\mathrm{diag}\left(1-(\boldsymbol{h}_{1,i}^\top\boldsymbol{a})^2, \ldots, 1-(\boldsymbol{h}_{K,i}^\top\boldsymbol{a})^2\right)\right)$$

$$= -\frac{2\tau}{N}\sum_{i=1}^{n}\boldsymbol{u}^\top\mathrm{diag}\left(1-(\boldsymbol{h}_{1,i}^\top\boldsymbol{a})^2, \ldots, 1-(\boldsymbol{h}_{K,i}^\top\boldsymbol{a})^2\right)\left(\boldsymbol{u} - \frac{K \bmod 2}{K}\mathbf{1}_K\right)$$

$$= -\frac{2\tau}{N}\sum_{i=1}^{n}\sum_{k=1}^{K}(1-(\boldsymbol{h}_{k,i}^\top\boldsymbol{a})^2)u_k^2 + (K \bmod 2)\frac{2\tau}{NK}\sum_{i=1}^{n}\sum_{k=1}^{K}(1-(\boldsymbol{h}_{k,i}^\top\boldsymbol{a})^2)u_k$$

$$\leq -\frac{2\tau}{N}\left(N - \|\boldsymbol{H}^\top\boldsymbol{a}\|_2^2\right) + (K \bmod 2)\frac{2\tau}{NK}\left(N - \|\boldsymbol{H}^\top\boldsymbol{a}\|_2^2\right)$$

$$= -\frac{2\tau}{NK}\left(N - \|\boldsymbol{H}^\top\boldsymbol{a}\|_2^2\right)\left(K - (K \bmod 2)\right).$$

Finally, the remaining term $-\tau\sum_{k=1}^{K}\alpha_k\|\boldsymbol{\delta}_{W_k}\|_2^2 - \tau\sum_{i=1}^{N}\beta_i\|\boldsymbol{\delta}_{H_i}\|_2^2$ in (31) vanishes, which is due to the fact that $\boldsymbol{\beta} = \mathbf{0}$ and

$$\sum_{k=1}^{K}\alpha_k\|\boldsymbol{\delta}_{W_k}\|_2^2 = \sum_{k=1}^{K}\alpha_k u_k^2 = \sum_{k=1}^{K}\alpha_k = 0,$$

where the last equality follows by Lemma 4 that $\sum_{k=1}^{K}\alpha_k = \sum_{i=1}^{N}\beta_i = 0$. Therefore, plugging both bounds above into (31), we obtain

$$\mathrm{Hess}\, f(\boldsymbol{W}, \boldsymbol{H})[\boldsymbol{\Delta}, \boldsymbol{\Delta}]$$

$$\leq \frac{\tau^2}{NK}\left(K - \frac{K \bmod 2}{K}\right)\|\boldsymbol{H}^\top\boldsymbol{a}\|_2^2 - \frac{2\tau}{NK}(N - \|\boldsymbol{H}^\top\boldsymbol{a}\|_2^2)(K - (K \bmod 2))$$

$$= \frac{\tau(K - (K \bmod 2))}{NK}\left(\tau\left[\frac{K^2 - (K \bmod 2)}{K(K - (K \bmod 2))}\right]\|\boldsymbol{H}^\top\boldsymbol{a}\|_2^2 - 2(N - \|\boldsymbol{H}^\top\boldsymbol{a}\|_2^2)\right)$$

$$= \frac{\tau(K - (K \bmod 2))}{NK}\left[(\tau[1 + (K \bmod 2)/K)] + 2)\|\boldsymbol{H}^\top\boldsymbol{a}\|_2^2 - 2N\right] < 0,$$

where the last inequality follows by our choice of $\boldsymbol{a} \in \mathbb{S}^{d-1}$ in Lemma 6. Thus we obtain the desired result in (32) for this case.

**Case 2:** Suppose $\beta_i \neq 0$ for all $i \in [N]$. Using the fact that $d > K$, choose $\boldsymbol{a} \in \mathbb{S}^{d-1}$ such that $\boldsymbol{W}^\top\boldsymbol{a} = \mathbf{0}$. By Lemma 2, given that $\boldsymbol{W}\boldsymbol{g}_i = \beta_i\boldsymbol{h}_i$ for all $i \in [N]$, we have

$$\boldsymbol{a}^\top\boldsymbol{W}\boldsymbol{g}_i = \beta_i\boldsymbol{a}^\top\boldsymbol{h}_i = 0, \quad \forall\, i \in [N].$$

Thus, as $\beta_i \neq 0$ for all $i \in [N]$, this simply implies that $\boldsymbol{H}^\top\boldsymbol{a} = \mathbf{0}$. Now using Lemma 3, for any non-optimal critical point $(\boldsymbol{W}, \boldsymbol{H})$, there exists at least one $k \in [K]$ or $i \in [N]$ such that either

$$\alpha_k > -\sqrt{n}\|\boldsymbol{G}\|, \quad \text{or} \quad \beta_i > -\|\boldsymbol{G}\|/\sqrt{n}. \tag{36}$$

Let $\boldsymbol{u}_1 \in \mathbb{R}^K$ and $\boldsymbol{v}_1 \in \mathbb{R}^N$ be the left and right unit singular vectors associated with the leading singular values of $\boldsymbol{G}$, respectively. In other words, we have

$$\boldsymbol{u}_1^\top\boldsymbol{G}\boldsymbol{v}_1 = \|\boldsymbol{G}\|. \tag{37}$$

By letting $\boldsymbol{u} = -\boldsymbol{u}_1/\sqrt[4]{n}$, $\boldsymbol{v} = \sqrt[4]{n}\boldsymbol{v}_1$, we construct the negative curvature direction as

$$\boldsymbol{\Delta} = (\boldsymbol{\Delta}_W, \boldsymbol{\Delta}_H) = \left(\boldsymbol{a}\boldsymbol{u}^\top, \ \boldsymbol{a}\boldsymbol{v}^\top\right). \tag{38}$$

Since $\boldsymbol{W}^\top \boldsymbol{a} = \boldsymbol{0}, \boldsymbol{H}^\top \boldsymbol{a} = \boldsymbol{0}$, we have

$$\boldsymbol{W}^\top \boldsymbol{\Delta}_H + \boldsymbol{\Delta}_W^\top \boldsymbol{H} \ = \ \boldsymbol{W}^\top \boldsymbol{a}\boldsymbol{v}^\top + \boldsymbol{u}\boldsymbol{a}^\top \boldsymbol{H} \ = \ \boldsymbol{0},$$

so that from (7) we have

$$\nabla^2 f(\boldsymbol{W}, \boldsymbol{H})[\boldsymbol{\Delta}, \boldsymbol{\Delta}] \ = \ \nabla^2 g(\boldsymbol{M}) \left[\tau\left(\boldsymbol{W}^\top \boldsymbol{\Delta}_H + \boldsymbol{\Delta}_W^\top \boldsymbol{H}\right), \tau\left(\boldsymbol{W}^\top \boldsymbol{\Delta}_H + \boldsymbol{\Delta}_W^\top \boldsymbol{H}\right)\right]$$
$$+ 2\tau \left\langle \boldsymbol{G}, \boldsymbol{\Delta}_W^\top \boldsymbol{\Delta}_H \right\rangle.$$

Thus, from (31), combining all the above derivations we obtain

$$\text{Hess} f(\boldsymbol{W}, \boldsymbol{H})[\boldsymbol{\Delta}, \boldsymbol{\Delta}] \ = \ 2\tau \left\langle \boldsymbol{G}, \boldsymbol{\Delta}_W^\top \boldsymbol{\Delta}_H \right\rangle - \tau \sum_{k=1}^K \alpha_k \|\boldsymbol{\delta}_{W_k}\|_2^2 - \tau \sum_{i=1}^N \beta_i \|\boldsymbol{\delta}_{H_i}\|_2^2.$$

$$= \ -2\tau \left\langle \boldsymbol{G}, \boldsymbol{u}_1 \boldsymbol{v}_1^\top \right\rangle - \tau \left(\sum_{k=1}^K \frac{\alpha_k u_{1,k}^2}{\sqrt{n}} + \sum_{i=1}^N \sqrt{n}\beta_i v_{1,i}^2\right)$$

$$= \ \tau \left(-2\|\boldsymbol{G}\| - \sum_{k=1}^K \frac{\alpha_k u_{1,k}^2}{\sqrt{n}} - \sum_{i=1}^N \sqrt{n}\beta_i v_{1,i}^2\right)$$

where the last equality follows from (37). On the other hand, by Lemma 3, the fact we derived in (36) that there exists $k \in [K]$ such that $\alpha_k > -\sqrt{n}\|\boldsymbol{G}\|$ or there exists $i \in [N]$ such that $\beta_i > -\|\boldsymbol{G}\|/\sqrt{n}$, and that $\|\boldsymbol{u}_1\|_2 = \|\boldsymbol{v}_1\|_2 = 1$, we obtain

$$-\sum_{k=1}^K \frac{\alpha_k u_{1,k}^2}{\sqrt{n}} - \sum_{i=1}^N \sqrt{n}\beta_i v_{1,i}^2 \ < \ \|\boldsymbol{G}\| \left(\sum_{k=1}^K u_{1,k}^2 + \sum_{i=1}^N v_{1,i}^2\right) \ = \ 2\|\boldsymbol{G}\|.$$

Therefore, we have

$$\text{Hess} f(\boldsymbol{W}, \boldsymbol{H})[\boldsymbol{\Delta}, \boldsymbol{\Delta}] \ < \ \tau\left(-2\|\boldsymbol{G}\| + 2\|\boldsymbol{G}\|\right) \ = \ 0,$$

as desired. □

*Proof of Theorem 2.* Let $(\boldsymbol{W}, \boldsymbol{H}) \in \mathcal{OB}(d, K) \times \mathcal{OB}(d, N)$ be a local minimizer of Problem (4). Suppose that it is not a global minimizer. This implies $(\boldsymbol{W}, \boldsymbol{H})$ is a critical point that is not a global minimizer. According to Proposition 2, the Riemannian Hessian at $(\boldsymbol{W}, \boldsymbol{H})$ has negative curvature. This contradicts with the fact that $(\boldsymbol{W}, \boldsymbol{H})$ is a local minimizer. Thus, we concludes that any local minimizer of Problem (4) is a global minimizer in Theorem 1. Moreover, according to Proposition 2, any critical point of Problem (4) that is not a local minimizer is a Riemmannian strict saddle point with negative curvature. □

# E  Experiments

In this section of the appendix, we provide details of the experimental setups, as well as additional experiments to support the main text.

**Network architectures, datasets, and training details.** In our experiments, we use ResNet [9] architectures for the feature encoder. For the normalized network, we project the output of the encoder onto the sphere of radius $\tau$ (as done in [7]) and also project the weight classifiers to the unit sphere after each optimization step to maintain constraints. In all experiments, we set $\tau = 1$. For the regularized UFM and network, we use a weight and feature decay of $10^{-4}$ (using the loss in [4]). We do not use a bias term for the classifier for either architecture. For all experiments, we use the CIFAR dataset[4] [12], where we use CIFAR100 for all experiments except for the experiments in Section 4.1 and Appendix E.3, where we use CIFAR10. In all experiments, we train the networks using SGD with a batch size of $128$ and momentum $0.9$ with an initial learning rate of $0.05$, and we decay the learning rate by a factor of $0.1$ after every $40$ epochs - these hyperparameters are chosen to be the same as those in [4] for fair comparisons. All networks are trained on Nvidia Tesla V100 GPUs with 16G of memory.

---

[4]Both CIFAR10 and CIFAR100 are publicly available and are licensed under the MIT license.

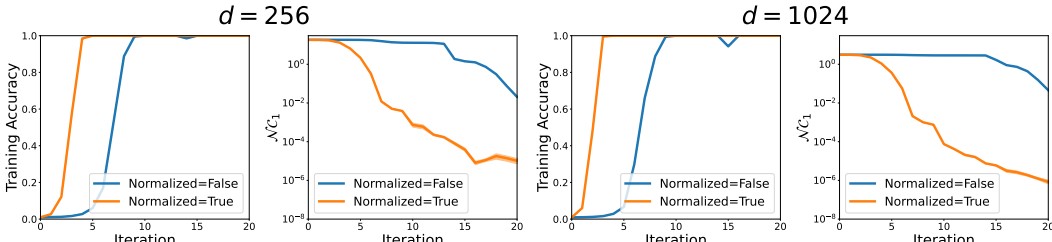

Figure 7: **Faster training/feature collapse of UFM with feature normalization with CG.** Average (deviation denoted by shaded area) training accuracy and $\mathcal{NC}_1$ of UFM over 10 trials of (Riemmanian) conjugate gradient method. We set $K = 100$ classes, $n = 30$ samples per class.

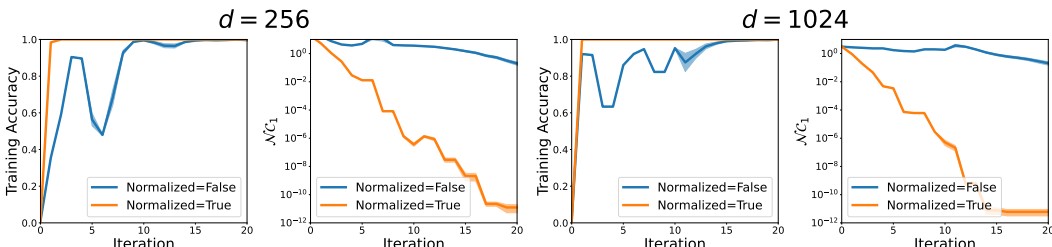

Figure 8: **Faster training/feature collapse of UFM with feature normalization with TRM.** Average (deviation denoted by shaded area) training accuracy and $\mathcal{NC}_1$ of UFM over 10 trials of (Riemmanian) trust-region method. We set $K = 100$ classes, $n = 30$ samples per class.

**Neural collapse metrics.** For measuring different aspects of neural collapse as introduced in Section 1, we adopt similar $\mathcal{NC}$ metrics from [1, 4, 5], given by

$$\mathcal{NC}_1 := \frac{1}{K}\text{trace}(\mathbf{\Sigma}_W \mathbf{\Sigma}_B^\dagger)$$

$$\mathcal{NC}_2 := \left\| \frac{\boldsymbol{W}^\top \boldsymbol{W}}{\|\boldsymbol{W}^\top \boldsymbol{W}\|_F} - \frac{1}{\sqrt{K-1}}\left(\boldsymbol{I}_K - \mathbf{1}_K \mathbf{1}_K^\top\right)\right\|_F$$

$$\mathcal{NC}_3 := \left\| \frac{\boldsymbol{W}^\top \overline{\boldsymbol{H}}}{\|\boldsymbol{W}^\top \overline{\boldsymbol{H}}\|_F} - \frac{1}{\sqrt{K-1}}\left(\boldsymbol{I}_K - \mathbf{1}_K \mathbf{1}_K^\top\right)\right\|_F,$$

where $\mathbf{\Sigma}_W$ and $\mathbf{\Sigma}_B$ are the within-class and between-class covariance matrices (see [1, 4] for more details), $\mathbf{\Sigma}_B^\dagger$ denotes pseudo inverse of $\mathbf{\Sigma}_B$, and $\overline{\boldsymbol{H}}$ is the centered class mean matrix in (9). More specifically, $\mathcal{NC}_1$ measures **NC1** (i.e., within class variability collapse), $\mathcal{NC}_2$ measures **NC2** (i.e., the convergence to the simplex ETF), and $\mathcal{NC}_3$ measures **NC3** (i.e., the duality collapse).

### E.1 Riemannian conjugate gradient and trust-region method for solving (4) under UFM

In Section 4, we demonstrated that optimizing Problem (4), which corresponds to the feature normalized UFM, results in quicker training and feature collapse as opposed to the regularized UFM formulation, as shown in Figure 5. To show that this phenomenon is independent of the algorithm used, we additionally test the conjugate gradient (CG) method [61] as well as the trust-region method (TRM) [61] to solve (4) with the same set-up as in Figure 5. While the Riemannian conjugate gradient method is also a first order method like gradient descent, the Riemannian trust-region method is a second order method, so the convergence speed is much faster compared to Riemannian gradient descent or conjugate gradient method. The results are shown in Figures 7 and 8 for the CG method and TRM respectively.

We see that optimizing the feature normalized UFM with CG gives similar results to using GD, whereas optimizing the feature normalized UFM using TRM results in an even greater gap in convergence speed to the global solutions, when compared with optimizing the regularized counterpart using TRM. These results suggest that the benefits of feature normalization are not limited to vanilla gradient descent or even first order methods.

Table 2: **Better generalization and test feature collapse with ResNet on CIFAR100 with feature normalization.** Test accuracy and test $\mathcal{NC}_1$ of ResNet-18 and ResNet-50 on CIFAR100.

|  | ResNet-18 | | ResNet-50 | |
| --- | --- | --- | --- | --- |
|  | **Test Accuracy** | **Test $\mathcal{NC}_1$** | **Test Accuracy** | **Test $\mathcal{NC}_1$** |
| Regularization | 55.3% | 3.838 | 48.9% | 4.486 |
| Normalization | **58.6%** | **3.143** | **56.4%** | **3.127** |

### E.2 Feature normalization generalizes better than regularization

In Section 4, we showed that using feature normalization over regularization improves training speed and feature collapse when training increasingly overparameterized ResNet models on a small subset of CIFAR100. We now demonstrate that feature normalization leads to better generalization than regularization.

We train a ResNet-18 and ResNet-50 model on the *entirety* of the CIFAR100 training split without any data augmentation for 100 epochs, and test the accuracy and $\mathcal{NC}_1$ metric on the standard test split. The results are shown in Table 2. We immediately see that using feature normalization gives both better test accuracy and test feature collapse than using regularization. Furthermore, the test generalization performance is coupled with the degree of feature collapse, supporting the claim that better $\mathcal{NC}$ often leads to better generalization performance. Finally, as we have trained both ResNet architectures with the same set-up and number of epochs, there is a substantial drop in performance (both test accuracy and test $\mathcal{NC}_1$) of the regularized ResNet-50 model compared to the ResNet-18. However, using feature normalization, this effect is mostly mitigated, suggesting that feature normalization is more robust compared to regularization and effective for generalizing highly overparameterized models on fixed-size datasets.

### E.3 Investigating the effect of the temperature parameter $\tau$

In Section 2, although the temperature parameter $\tau > 0$ does not affect the global optimality and critical points, it does affect the training speed of specific learning algorithms and hence test performance. In all experiments in Section 4, we set $\tau = 1$ and have not discussed in detail the effects of $\tau$ in practice.

However, as mentioned in [7], $\tau$ has important side-effects on optimization dynamics and must be carefully tuned in practice. Hence, we now present a brief study of the temperature parameter $\tau$ when optimizing the problem (4) under the UFM and training a deep network in practice. To begin, we consider the UFM formulation. We first note that $\tau$ does not affect the theoretical global solution or benign landscape of the UFM (although it does affect the attained theoretical lower bound, see Theorem 1 and Figure 3). However, it does impact the rate of convergence to neural collapse as well as the attained numerical values of the $\mathcal{NC}$ metrics. To see this, we apply (Riemannian) gradient descent with backtracking line search to Problem (4) for various settings of $\tau$. These results are shown in Figure 9. First, it is evident that for all tested $\tau$ values, we achieve perfect classification in a similar number of iterations. Furthermore, the rate of convergence of $\mathcal{NC}_1$ is somewhat the same for most settings of $\tau$, and we essentially have feature collapse for most settings of $\tau$. On the other hand, it appears that the rate of convergence of $\mathcal{NC}_2$ and $\mathcal{NC}_3$ are dramatically affected by $\tau$, with values in the range of 1 to 10 yielding the greatest collapse. This aligns with the choice of the temperature parameter in the experimental section of [7], where the equivalent parameter is set $\rho = 1/\sqrt{0.1} \approx 3.16$.

We now look to the setting of training practical deep networks. We train a feature normalized ResNet-18 architecture on CIFAR-10 for various settings of $\tau$. The results are shown in Figure 10. One immediate difference from the UFM formulation is that we arrive at perfect classification of the training data for a particular range of values for $\tau$ (from about 0.1 to 10) but not for all settings of $\tau$. Within this range, we can see that values of $\tau$ around 1 to 10 lead to the fastest training, and values close to $\tau = 5$ lead to the greatest collapse in all $\mathcal{NC}$ metrics, as was the case with the UFM. All this evidence suggests that $\tau = 1$ is not the optimal setting of the temperature parameter for either

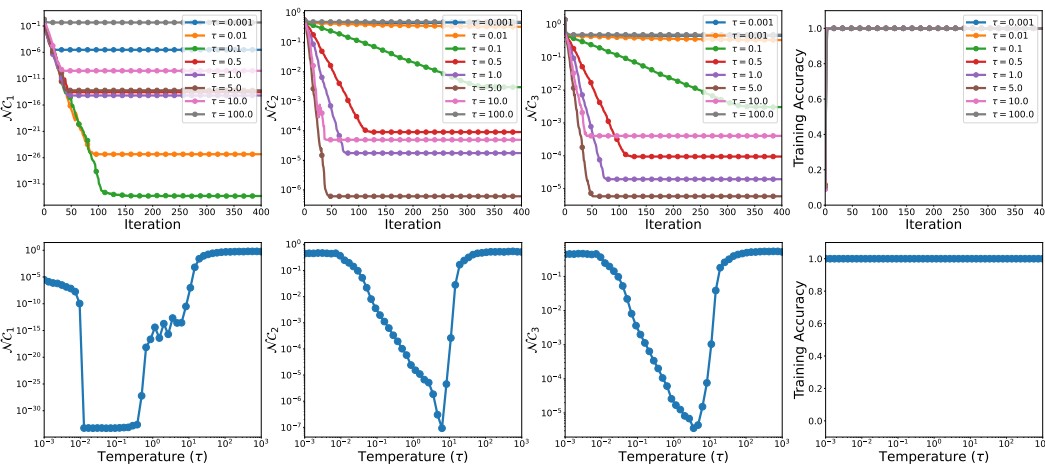

Figure 9: **Effect of temperature parameter on collapse and training accuracy of UFM.** $K = 10$ classes, $n = 5$ samples per class, $d = 32$. Top row: Average $\mathcal{NC}$ metrics and training accuracy of UFM over 20 trials for various settings of $\tau$ with respect to each iteration of (Riemannian) gradient descent. Bottom row: Final average $\mathcal{NC}$ metrics and training accuracy of UFM over 20 trials for various settings of $\tau$.

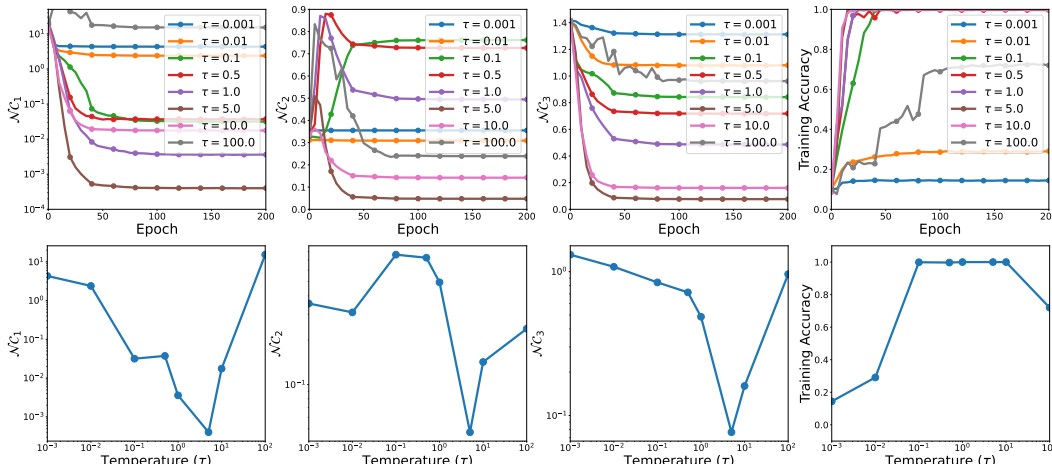

Figure 10: **Effect of temperature parameter on collapse and training accuracy of ResNet.** Top row: Average $\mathcal{NC}$ metrics and training accuracy of ResNet-18 on CIFAR-10 with $n = 100$ for various settings of $\tau$ with respect to each epoch over 200 epochs. Bottom row: Final average $\mathcal{NC}$ metrics and training accuracy of ResNet-18 on CIFAR-10 with $n = 100$ for various settings of $\tau$.

the UFM or ResNet, particularly when measuring $\mathcal{NC}_2$ and $\mathcal{NC}_3$, and instead $\tau = 5$ may perform better. In the practical experiments of the main text, however, we mainly focused on training speed and feature collapse, and for these purposes it appears that the $\tau$ parameter can be set in a fairly nonstringent manner.

## E.4 Benign global landscape of other classification losses

Although the cross-entropy (CE) loss studied in this work is arguably the most common loss function for deep classification tasks, it is not the only one. Some other commonly used loss functions include focal loss (FL) [83], label smoothing (LS) [84], and supervised contrastive (SC) loss [34], each of which has demonstrated various benefits over vanilla CE. In this section, we briefly explore the empirical global landscape of these losses under the UFM with normalized features (and classifiers).

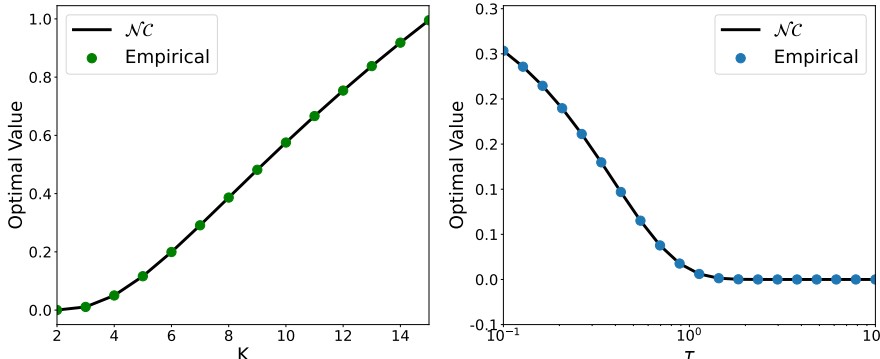

Figure 11: **Global optimization of focal loss (FL) with $\gamma = 3$ under UFM with $d = 16$ and $n = 3$.** Black line refers to theoretical value of (39) at $\mathcal{NC}$ solutions. Empirical values found using gradient descent with random initialization. Left: Lower bound against number of classes $K$ while fixing $\tau = 1$. Right: Lower bound against temperature $\tau$ while fixing $K = 3$. The same empirical values are achieved over many trials.

Specifically, we consider the problem

$$\min_{\boldsymbol{W},\boldsymbol{H}} f(\boldsymbol{W},\boldsymbol{H}) := \frac{1}{N} \sum_{k=1}^{K} \sum_{i=1}^{n} \mathcal{L}\left(\tau \boldsymbol{W}^\top \boldsymbol{h}_{k,i}, \boldsymbol{y}_k\right), \tag{39}$$

$$\text{s.t.} \quad \boldsymbol{H} \in \mathcal{OB}(d,N), \ \boldsymbol{W} \in \mathcal{OB}(d,K).$$

where $\mathcal{L}$ is either the focal loss or label smoothing loss.

First, we consider the focal loss, defined as

$$\mathcal{L}_{\mathrm{FL}}(\boldsymbol{z}, \boldsymbol{y}_k) = -\left(1 - \frac{\exp(z_k)}{\sum_{\ell=1}^{K} \exp(z_\ell)}\right)^\gamma \log\left(\frac{\exp(z_k)}{\sum_{\ell=1}^{K} \exp(z_\ell)}\right)$$

where $\gamma \geq 0$ is the *focusing* parameter (with $\gamma = 0$, we recover ordinary CE). As seen in Figure 11, using gradient descent with random initialization on the focal loss, we achieve neural collapse over a range of settings of $K$ and $\tau$. Characterizing the global solutions of (39) for the focal loss and a general landscape analysis are left as future work.

Next, we consider the label smoothing loss, defined as

$$\mathcal{L}_{\mathrm{LS}}(\boldsymbol{z}, \boldsymbol{y}_k) = -\left(1 - \frac{K-1}{K}\alpha\right) \log\left(\frac{\exp(z_k)}{\sum_{\ell=1}^{K} \exp(z_\ell)}\right) - \frac{\alpha}{K} \sum_{j \neq k} \log\left(\frac{\exp(z_j)}{\sum_{\ell=1}^{K} \exp(z_\ell)}\right)$$

where $\alpha \geq 0$ is the *smoothing* parameter (with $\alpha = 0$, we recover ordinary CE). As seen in Figure 12, for small enough $\tau$ we achieve neural collapse, but for larger $\tau$, we do not. In fact, the global solutions of the label smoothing loss are not neural collapse for large enough $\tau$. To see this, let $(\boldsymbol{W}_1, \boldsymbol{H}_1)$ denote a $\mathcal{NC}$ solution, and let $(\boldsymbol{W}_2, \boldsymbol{H}_2)$ denote a solution where $\boldsymbol{W}_2 = \boldsymbol{a}\boldsymbol{1}_K^\top$ and $\boldsymbol{H}_2 = \boldsymbol{a}\boldsymbol{1}_N^\top$, where $\boldsymbol{a}$ is unit-norm. It is easy to compute that

$$f(\boldsymbol{W}_1, \boldsymbol{H}_1) = \log\left(1 + (K-1)\exp\left(-\frac{K\tau}{K-1}\right)\right) + \alpha\tau$$

so $f(\boldsymbol{W}_1, \boldsymbol{H}_1) \to \infty$ as $\tau \to \infty$, whereas $f(\boldsymbol{W}_2, \boldsymbol{H}_2) = \log(K)$ is independent of $\tau$. Again, characterizing the global solutions of (39) for label smoothing and a general landscape analysis are left as future work.

Finally, we look to the supervised contrastive loss. Unlike the other losses, we do not have

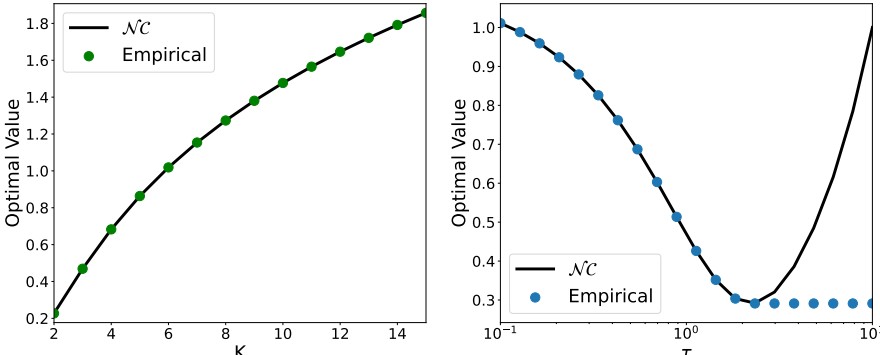

Figure 12: **Global optimization of label smoothing (LS) with** $\alpha = 0.1$ **under UFM with** $d = 16$ **and** $n = 3$. Black line refers to value of (39) at $\mathcal{NC}$ solutions. Empirical values found using gradient descent with random initialization. Left: Lower bound against number of classes $K$ while fixing $\tau = 1$. Right: Lower bound against temperature $\tau$ while fixing $K = 3$. The same empirical values are achieved over many trials.

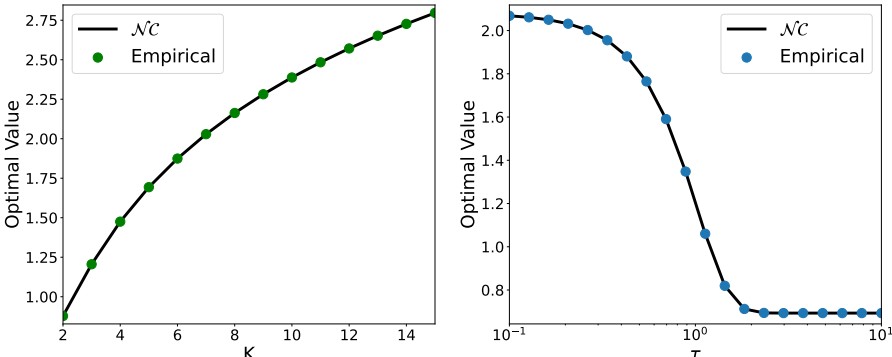

Figure 13: **Global optimization of supervised contrastive (SC) loss under UFM with** $d = 16$ **and** $n = 3$. Black line refers to value of (40) at $\mathcal{NC}$ solutions. Empirical values found using gradient descent with random initialization. Left: Lower bound against number of classes $K$ while fixing $\tau = 1$. Right: Lower bound against temperature $\tau$ while fixing $K = 3$. The same empirical values are achieved over many trials.

classifier $\boldsymbol{W}$ when training, so we instead have the problem

$$\min_{\boldsymbol{H}} f(\boldsymbol{H}) := -\frac{1}{N(n-1)} \sum_{i=1}^{N} \sum_{\substack{j \neq i \\ y_j = y_i}} \log\left(\frac{\exp(\tau^2 \boldsymbol{h}_i^\top \boldsymbol{h}_j)}{\sum_{\ell \neq i} \exp(\tau^2 \boldsymbol{h}_i^\top \boldsymbol{h}_\ell)}\right) \tag{40}$$
$$\text{s.t.} \quad \boldsymbol{H} \in \mathcal{OB}(d, N).$$

We note that the loss as written above computes the loss over the entire dataset, as opposed to computing over all minibatches of a fixed size as in [7]. As seen in Figure 13, using gradient descent with random initialization on the supervised contrastive loss, we achieve neural collapse over a range of settings of $K$ and $\tau$. In fact, it is proven in [7] that the global minimizers of (40) are $\mathcal{NC}$ solutions. However, an understanding of the global landscape requires further exploration and is left as future work.