# OpenReview forum: "Neural Collapse with Normalized Features: A Geometric Analysis over the Riemannian Manifold"
_NeurIPS.cc/2022/Conference — NeurIPS 2022 Accept_

### Official Review · Reviewer_VCm4 · 2022-07-11

**Rating:** 6
**Confidence:** 4
**Soundness:** 2 fair
**Presentation:** 3 good
**Contribution:** 3 good

**Summary:**

The paper provides theoretical justification of the neural collapse phenomenon for normalized features. The proof is based on the unconstrained feature model (UFM) and the resulting optimization problem is analyzed over the Riemannian manifold by constraining features and classifiers prototypes over the sphere. The paper basically shows that critical points for the optimization (i.e. potential minima) are saddles (i.e. negative curvature critical points) while all others are global minimizers for which the neural collapse properties hold. The paper concludes with an empirical evaluation confirming the proposed theory.


**Questions:**

This form is intentionally left blank as questions are included in the previous form.


**Limitations:**

No negative societal impacts. Limitations are included into the Strengths And Weaknesses form.

**Strengths And Weaknesses:**

STRENGTHS:

1) Interesting formulation of neural collapse in the case in which features are normalized. Feature normalization is a quite classic strategy/practice that has been used in several deep learning contexts (i. e. face recognition) in which maximal separation between learned features is required and therefore it naturally matches with the concept of maximal separation which is inherent to the neural collapse phenomenon. The provided theoretical analysis is very useful. The fact that a standard practice has a convergent geometric discriminative structure is a clear strength of this paper.

2) The oblique manifold formulation is interesting and it appears to be well suited to the problem.

WEAKNESSES:

1) The paper is well written, but many parts seem to require further work. The main result (the proof according to the oblique manifold formulation) of the paper is neither introduced nor motivated and no intuitions are given. Although very interesting, it is hard for the reviewer and, possibly for the reader, to fully grasp the real contribution of the paper. The paper refers to a book (specifically to some exercises) from which the reader should possibly get some intuition and motivations. This part should be “softened”.

2) The related work section in its current shape is somewhat short and without specific comments. It should be improved by following a thread with respect to what is proposed. The reviewer agrees that feature normalization can improve separability, however the four cited works (i.e. [22,23,24,7]) are not discussed in this regard. In which way those four works achieve separability? What is the quality of the representation? Does it depend on the task addressed? Is there any trade-off in the achievement of separability? Important related works are also missing. Many papers, especially from the feature learning and visual search literature are completely missing [A,B,C,D,E,F]. In particular it seems that [A] is one the first works which studied classifier prototypes and their relationship with the learned features. The works [B,C] and [D] firstly introduced feature normalization and their feature separability (i.e., discriminative features) behavior. Finally, [E] and [F] seem to be the first papers to apply maximal separability in a simplex shaped structure (and other regular polytopes). As recent neural collapse literature used the term simplex ETF, what is the difference between the  simplex regular polytopes and a simple ETF? Are they equivalent?

3) The introduction should be better focused on the related work. On line 37 a large number of works are cited in a single shot as related to NC. However no comments are explicitly given. Given the diversity of the cited papers a discussion should be given.

4) What data is used in Fig.1 and Tab.1? It seems that it is not real data. How is the data generated?

5) The most important part related to the proof, should be improved. Some motivations and intuitions about the oblique manifold should be given. The cited book is addressing a dictionary learning problem. Although the affinity is clear, the paper in its current shape does not provide an intuitive access to the oblique manifold formulation. A potential reader of this paper would be forced to read too many references. The reviewer's suggestion is to increase the level of self-containment of the paper.

References

[A] Liu, W., Wen, Y., Yu, Z., & Yang, M. (2016). Large-margin softmax loss for convolutional neural networks. arXiv preprint arXiv:1612.02295.

[B] Ranjan, R., Castillo, C. D., & Chellappa, R. (2017). L2-constrained softmax loss for discriminative face verification. arXiv preprint arXiv:1703.09507.

[C] Wang, F., Xiang, X., Cheng, J., & Yuille, A. L. (2017, October). Normface: L2 hypersphere embedding for face verification. In Proceedings of the 25th ACM international conference on Multimedia (pp. 1041-1049).

[D] Liu, W., Wen, Y., Yu, Z., Li, M., Raj, B., & Song, L. (2017). Sphereface: Deep hypersphere embedding for face recognition. In Proceedings of the IEEE conference on computer vision and pattern recognition (pp. 212-220).

[E] Pernici, F., Bruni, M., Baecchi, C., & Del Bimbo, A. (2019, January). Maximally Compact and Separated Features with Regular Polytope Networks. In CVPR Workshops (pp. 46-53).

[F] Pernici, F., Bruni, M., Baecchi, C., & Del Bimbo, A. (2021). Regular polytope networks. IEEE Transactions on Neural Networks and Learning Systems.

---

> ### Author Response · Authors · 2022-08-02
> **Response to Reviewer VCm4**
>
> We thank the reviewer for their helpful comments. Please find our response below.
>
> ${\bf Q1.}$ The main result of the paper is neither introduced nor motivated and no intuitions ... This part should be "softened".  (see Weakness 1 for details)
>
> ${\bf A1.}$ We thank the reviewer for valuable feedback and suggestions. In the revision, we have provided more motivations and intuitions about our main results by connecting feature normalization to discriminative representations (see the sentences in blue on Page 2 and the discussions in Section A of the appendix). We have also added more details about the calculus on the oblique manifold in Section 2.2. More specifically, we have added Figure 2 to illustrate the tangent space and Riemannian gradient of a reduced oblique manifold, with some sentences in blue for providing more intuitions behind the derivation. Additionally, we provided more technical details for deriving Eq. (6) and (7) in Appendix B.2.
>
>
> ${\bf Q2.}$ The related work section in its current shape is somewhat short and without specific comments. It should be ... As recent neural collapse literature used the term simplex ETF, what is the difference between the simplex regular polytopes and the simplex ETF? Are they equivalent?  (see Weakness 2 for details)
>
> ${\bf A2.}$ We have modified our manuscript according to your valuable comments. We have discussed our cited work [7,22,23,24] ([7,30,31,32] in the revised manuscript) in Section A of the appendix as suggested. Moreover, we also cited and discussed [A,B,C,D,E,F] ([15,16,17,18,21,22] in the revised manuscript) in Section A of the appendix.
>
> The simplex regular polytopes and the simplex ETF are not equivalent. Indeed, we suppose that your mentioned simplex regular polytopes refers to the regular simplex in [E,F], which is a simplex and a regular polytope. In our Theorem 1, the simplex ETF refers to a collection of vectors $\{h_k\}_{k=1}^K \subseteq \mathbb{R}^d$ satisfying $\|h_k\|=1$ and $h_k^Th_\ell=-\frac{1}{K-1}$ for all $k\neq \ell$. Then, we cannot say the simplex ETF is a regular simplex. However, we can say the $\textbf{convex hull}$ formed by the simplex ETFs and the regular simplex are equivalent if $K=d+1$.
>
>
> ${\bf Q3.}$ The introduction should be better focused on the related work.  (see Weakness 3 for details)
>
> ${\bf A3.}$ We have added the discussion of the cited work in Line 37 in Appendix A in the revised manuscript.
>
>
> ${\bf Q4.}$ What data is used in Fig. 1 and Tab. 1? It seems that it is not real data. How is the data generated?  (see Weakness 4 for details)
>
> ${\bf A4.}$ In Figure 1 and Table 1, we do not use any real data. Instead, we use the unconstrained feature model (UFM), meaning the resulting features are independent of any input data, we simply pair each $h_{k,i}$ with the corresponding one-hot label $y_k$. For the normalized version, we optimize Problem (3) with $K=100$ and $n=5$ to find the features and classifiers, whereas for the non-normalized version, we optimize Problem (3) without the manifold constraint, i.e., the features and classifiers are not normalized. To make it clear that the features are not derived from data, we have changed the word "learned" to "found" in the caption of Figure 1.
>
>
> ${\bf Q5.}$ The most important part related to the proof should be ... is to increase the level of self-containment of the paper.  (see Weakness 5 for details)
>
> ${\bf A5.}$ As suggested, we have improved our proof part in Section 2.2 (see the sentences in blue and Appendix B.2) and provided more motivations and intuitions about the oblique manifold (see Figure 2 and the sentences in blue).

---

### Official Review · Reviewer_vkF5 · 2022-07-11

**Rating:** 6
**Confidence:** 2
**Soundness:** 3 good
**Presentation:** 3 good
**Contribution:** 3 good

**Summary:**

The paper considers the landscape of the optimization problem when training neural networks with normalized features under the cross-entropy loss. Under the assumption of the unconstrained feature model, the authors reformulated the problem as a Riemannian optimization problem, and showed that (1) global solutions of the problem satisfy the neural collapse properties and (2) the optimization problem has a benign global landscape (i.e. local minimizers are global minimizers, and all other critical points are strict saddle points).

**Questions:**

This is probably a minor question regarding the presentation of Figure 2.

In the end of section 3, the "Limitations on the assumptions of the feature dimension d" part, the authors stated that "We believe the bound can be improved (from d>N) to d>K+1, .... corroborated by our experimental results (See Figure 2)".
 - I am expecting Figure 2 to show an experiment with K+1<d<N, which would empirically support the author's claims that we do not necessarily need d>N.
 - But instead, it looks like the authors chose d=100, n=5 for that Figure, which still guarantees d>n, so it does not really corroborate the claim above. Am I missing something, or should the graph be modified?

**Strengths And Weaknesses:**

__Quality and Clarity__: The paper is written clearly, and I enjoyed reading the motivations that builds up nicely to the main theorems. The experiments also support most of the author's claims well.

__Originality and Significance__: I personally do not work in this field, so I will only provide some brief observations on significance of results, while leaving more discussions to other expert reviewers: The proof of theorem 1 on neural collapse properties is not too different from existing techniques cited by this paper. Theorem 2, on the other hand, is nontrivial, and understanding the global landscape is indeed useful for developing guarantees for algorithms like Riemannian SGD.

---

> ### Author Response · Authors · 2022-08-02
> **Response to Reviewer vkF5**
>
> We thank the reviewer for their helpful comments. Please find our response below.
>
> ${\bf Q1.}$ The proof of Theorem 1 on neural collapse properties is not different from the existing techniques cited by this paper.
>
> ${\bf A1.}$ We agree with the reviewer that our major contribution lies in Theorem 2 instead of Theorem 1. Nonetheless, we want to mention that our proof of Theorem 1 is different from that in [4,5]. Indeed, we first reduce Problem (3) with $nK$ variables $h_{k,i}$ for all $k \in [K],i \in [n]$ into Problem (12) with $K$ variables $h_k$ for $k\in [K]$ in Lemma 1 of Appendix B. Then, we just need to analyze the optimal solution of Problem (12). This technique greatly simplifies our analysis and has not been used in [4,5]. We also refer the reviewer to our response A1 to Reviewer V69c for extra comments.
>
> [4] Zhihui Zhu, Tianyu Ding, Jinxin Zhou, Xiao Li, Chong You, Jeremias Sulam, and Qing Qu. A geometric analysis of neural collapse with unconstrained features. Advances in Neural Information Processing Systems, 34, 2021.
>
> [5] Jinxin Zhou, Xiao Li, Tianyu Ding, Chong You, Qing Qu, and Zhihui Zhu. On the optimization landscape of neural collapse under mse loss: Global optimality with unconstrained features. arXiv preprint arXiv:2203.01238, 2022.
>
> ${\bf Q2.}$ Issue of Figure 2
>
> ${\bf A2.}$ First, it is worth mentioning that we have improved our bound from $d>N$ to $d>K$ in Theorem 2 in the revised manuscript by a more delicate analysis. Thus, for $d=100$ and $n=5$ in Figure 2 (Figure 3 in the revised version), we always have $K < d$ for $K \in [5,50]$  so that the the condition for Theorem 2 holds. Additionally, we have  $K<d<N$ (Note that $N=nK$ in our paper) for $K > 20$,  and we have plotted results for $K$ up to $K=50$.

---

### Official Review · Reviewer_V69c · 2022-07-17

**Rating:** 5
**Confidence:** 3
**Soundness:** 3 good
**Presentation:** 3 good
**Contribution:** 2 fair

**Summary:**

Feature normalization has been a common practice. In order to justify neural collapse for normalized features, this paper formulate an unconstrained feature model (UFM) with spherical constraints. Theoretical results on the neural collapse global optimality and the benign landscape are performed. Experiments are conducted to show that UFM with feature normalization has better training and collapse performances than feature regularization.

**Questions:**

For practical application, one is more interested in generalization ability. Regularization has been known as an effective scheme to improve generalization. It is not clear about this for feature normalization. So, I think the authors should compare validation/test accuracy in Figure 4 and 5.

**Limitations:**

There are discussions about the limitations of this work.

No concern about negative social impact.


**Strengths And Weaknesses:**

Strengths:

-	The motivation to interpret feature normalization from the neural collapse perspective is interesting.

-	The paper is well organized and presented.

-	Theoretical work is sound.

Weaknesses:

-	Limited contribution. The first theoretical result, the neural collapse global optimality of Eq. (3), is a minor extension of Theorem 1 in [1], which also has a spherical constraint. Besides, as stated by the author, the problem with spherical constraint in Eq. (3) has the same global solution as the one with ball constraint in [2,3]. But the feasible region of ball constraint in [2,3] is much larger. So, the result in this paper is less significant.

-	In order to verify the theoretical results (Theorem 1 and 2) in this paper, the authors show that UFM with feature normalization has faster training and better collapse than feature regularization. But, the same theoretical results, including the same global optimality of neural collapse conditions and the same benign landscape properties, are also applicable to UFM with regularization [4], and UFM without either constraint or regularization [5]. In this case, how to explain the advantage of normalization over regularization in the experiments?

[1] Lu et al., Neural collapse with cross-entropy loss,

[2] Graf et al., Dissecting supervised contrastive learning, ICML 2021,

[3] Fang et al., Exploring deep neural networks via layer-peeled model: Minority collapse in imbalanced training, PNAS,

[4] Zhu et al., A geometric analysis of neural collapse with unconstrained features, NeurIPS 2021,

[5] Ji et al., An unconstrained layer-peeled perspective on neural collapse,

---

> ### Author Response · Authors · 2022-08-02
> **Response to Reviewer V69c**
>
> We thank the reviewer for their insightful comments. Please find our response below.
>
> ${\bf Q1.}$ Limited contribution: (1) minor extension of Theorem 1 in [1]; (2) sphere constraint vs. ball constraint in [2,3]
>
> ${\bf A1.}$ First of all, we want to emphasize that our main theoretical contribution lies in Theorem 2, showing that the manifold formulation (3) has a benign global landscape, with proof in Appendix D. We made this clear at the bottom of page 6 during the revision. Additionally, we make the following clarifications for Theorem 1.
> * Compared to Theorem 1 in [1], our Theorem 1 is closer to the practical setting and more general due to the fact that [1] assumes that each class only has one sample $(n=1)$. In view of this, Theorem 1 in [1] cannot demonstrate (NC1) variability collapse.
> * Although the feasible region of the sphere constraint is smaller than that of the ball constraint, the sphere constraint is more widely-used in practice due to the fact that the feature normalization is a common technique in deep learning. Then, it is more reasonable and meaningful to study the sphere constraint. In addition, our Theorem 1 and those in [2,3] can imply each other. It is easy to see the theorems in [2,3] imply our Theorem 1. On the other hand, ours implies theirs due to the inequality
> $(1+c_1)(K-1)(\bar{f}(W,Q)-c_2) \ge -\frac{\tau}{2}(c_3\sum_{k=1}^K ||w_k||^2_2 + \frac{1}{c_3}\sum_{k=1}^K ||q_k||_2^2)$ in the proof of Proposition 1 in Appendix C.
>
> ${\bf Q2.}$ Explain the advantage of normalization over regularization theoretically and experimental.
>
> ${\bf A2.}$ We agree with the reviewer that both feature normalization and regularization have benign landscape and the same global NC solution under UFM. Like the results in [4,5], as we only characterize the critical points, our current results are limited as it does not directly lead to polynomial convergence for iterative algorithms like GD or Riemannian GD. Indeed, even with benign landscape, in the worst case GD may take exponentially time to converge [A]. However, we conjecture the actual nonconvex landscape is much more benign than the worst-case cooked up in [A], and the local landscape around global NC solutions has certain regularity condition. Based upon our experiments in Figure 5 & 6, we conjecture that the feature normalization leads to better local regularity condition than that  of the regularization methods. We leave the thorough analysis as future work. We think one approach is to quantitively characterize and compare the regularity condition between the two approaches, and we can also provide more evidence through visualization like what has been done in [5,Figure 2].
>
> ${\bf Q3.}$ Compare validation/test accuracy in Figures 4 and 5
>
> ${\bf A3.}$ We thank the reviewer for the valuable question. In Appendix E.2, we have conducted new experiments and reported generalization performance of feature normalization vs. regularization in Table 2, showing that feature normalization gives better test accuracy and feature collapse for ResNet models on CIFAR100.
>
> [A] Du, Simon S., et al. "Gradient descent can take exponential time to escape saddle points." Advances in neural information processing systems 30 (2017).

---

### Author Response · Authors · 2022-08-02
**Response to all reviewers**

We express our gratitude to all reviewers for their insightful comments. Before addressing their comments, we would like to highlight some improvements that we made during the revision, where all major changes to the manuscript are highlighted in blue in both main body and supplementary materials.

(1) We improved the dimension bound in Theorem 2 from $d>N=nK$, which is quite loose, to $d>K$. The improved bound is based upon a tighter analysis in Appendix D.3.

(2) We improved the presentation of the paper, by providing more intuition on Riemannian calculus on page 5, and more comprehensive discussion on related work in Appendix A.

(3) We provided additional experimental results regarding better generalization properties of feature normalization in Appendix E.2, and we also conduct exploratory experiments for empirically demonstrating benign global landscapes of other commonly-used losses in Appendix E.4.

---

### Meta-Review · Area_Chair_9Vax · 2022-08-22

**Recommendation:** Accept
**Confidence:** Less certain

**Metareview:**

The paper studies a matrix decomposition problem and shows the problem is in a strict-saddle type. All the reviewers tend to accept the paper. I recommend an acceptance.

**Award:**

No

---

### Decision · Program_Chairs · 2022-09-14

Accept